# A hydrogen sulphide-responsive and depleting nanoplatform for cancer photodynamic therapy

Yuqi Zhang [1], Jing Fang[1], Shuyue Ye[1], Yan Zhao[1], Anna Wang[1], Qiulian Mao[1], Chaoxiang Cui[1], Yali Feng[1], Jiachen Li[1], Sunao Li [2], Mingyang Zhang [2] & Haibin Shi [1]✉

Hydrogen sulfide ($H_2S$) as an important biological gasotransmitter plays a pivotal role in many physiological and pathological processes. The sensitive and quantitative detection of $H_2S$ level is therefore crucial for precise diagnosis and prognosis evaluation of various diseases but remains a huge challenge due to the lack of accurate and reliable analytical methods in vivo. In this work, we report a smart, $H_2S$-responsive and depleting nanoplatform (ZNNPs) for quantitative and real-time imaging of endogenous $H_2S$ for early diagnosis and treatment of $H_2S$-associated diseases. We show that ZNNPs exhibit unexpected NIR conversion ($F_{1070} \rightarrow F_{720}$) and ratiometric photoacoustic ($PA_{680}/PA_{900}$) signal responsiveness towards $H_2S$, allowing for sensitive and quantitative visualization of $H_2S$ in acute hepatotoxicity, cerebral hemorrhage model as well as colorectal tumors in living mice. ZNNPs@FA simultaneously scavenges the mitochondrial $H_2S$ in tumors leading to significant ATP reduction and severe mitochondrial damage, together with the activated photodynamic effect, resulting in efficient suppression of colorectal tumor growth in mice. We believe that this platform may provide a powerful tool for studying the vital impacts of $H_2S$ in related diseases.

[1] State Key Laboratory of Radiation Medicine and Protection, School for Radiological and Interdisciplinary Sciences (RAD-X) and Collaborative Innovation Center of Radiation Medicine of Jiangsu Higher Education Institutions, Soochow University, Suzhou, Jiangsu, China. [2] Department of Forensic Sciences, School of Basic Medicine and Biological Sciences, Soochow University, Suzhou, China. ✉email: hbshi@suda.edu.cn

ntracellular hydrogen sulfide ($H_2S$), derived from L-Cysteinein-related enzymatic biosynthesis, has been well demonstrated to be an important gasotransmitter that is closely implicated in various physiological and pathological processes[1–4]. The low expression level of $H_2S$ is widely considered to have a valid effect on cytoprotective pathways due to its antioxidation, anti-apoptosis, and anti-inflammation[5–8]. However, the abnormal production of $H_2S$ in the living system was found to be essentially associated with the occurrence of many diseases, such as Alzheimer's disease[9,10], liver cirrhosis[11], inflammation[12,13], and cancers[14,15]. Therefore, accurately monitoring and depleting the endogenous $H_2S$ as a therapeutic target in the living system is of tremendous significance on early diagnosis and treatment of $H_2S$-related diseases.

To date, the most commonly used methods for detection of $H_2S$ include high-pressure liquid chromatography[16], gas chromatography[17], electrochemical analysis[18], and methylene blue assay[19], etc. These approaches display great detecting sensitivity in the order of nanomole. Unfortunately, the high cost, tedious operation, and low temporal resolution severely hinder their practical application for the detection of $H_2S$ in living subjects. To this end, great efforts have recently been devoted to exploring the intelligent $H_2S$-responsive probes for the diagnosis and therapy of various diseases. Numerous $H_2S$-responsive fluorescent probes have been developed on the basis of various mechanisms, for example, nucleophilic addition[20–22], copper sulfide precipitation[23], reduction of azido or nitro group[24], and thiolysis of dinitrophenyl ether[25], etc[26]. These probes have been demonstrated substantial advantages for $H_2S$ detection in biological systems, offering a powerful tool for real-time imaging of $H_2S$ due to quick response, noninvasiveness, and high sensitivity[27–29]. Nevertheless, most of them are technically challenge to apply in vivo due to the shallow tissue penetration, poor spatial resolution, and severe photon scattering. Therefore, it is highly meaningful to develop new strategies for accurate and quantitative assessment of $H_2S$ level in deep tissues for diagnosis and therapeutic evaluation of $H_2S$-associated diseases.

Photoacoustic (PA) imaging that synergistically integrates optical and ultrasound imaging overcomes the limitation of conventional fluorescence imaging interfered by the strong light absorption and scattering of tissues, and offers a promising imaging modality with deep tissue penetration capability and high spatial resolution[30–33]. On account of these significant advantages, a number of PA probes have been exploited for the detection of various analytes, such as reactive oxygen species (ROS)[34,35], pH[36], enzyme[37,38], metal ion[39], and $H_2S$[40]. However, the in vivo PA imaging of many physiological and pathological processes based on single responsive PA signal usually suffers from serious influence from the concentration of probes, environmental condition, and light scattering of tissues. Fortunately, ratiometric PA imaging that can overcome these shortcomings by self-calibration using two separated wavelengths of responsive PA signals to improve the signal-to-noise ratio of PA imaging has aroused a tremendous attention in bioimaging application. Although several ratiometric PA probes for $H_2S$ detection have been recently reported[40,41], the complicated fabrication, small strokes shift, and single functionality hinder their broad application in vivo. Hence, developing superior $H_2S$-responsive ratiometric PA imaging probes with quantitative imaging and therapeutic function has been regarded as a promising strategy to achieve accurate diagnosis and effective treatment of $H_2S$-related diseases.

In this work, we report a smart, $H_2S$-responsive and depleting nanoplatform ZNNPs to quantitatively visualize and simultaneously scavenge the endogenous $H_2S$, together with the activated photodynamic effect, for colorectal cancer treatment. As illustrated in Fig. 1, a synthesized hydrophobic NIR-II fluorophore ZM1068-NB as $H_2S$-responsive unit is encapsulated by biocompatible and amphiphilic mPEG$_{5000}$-PCL$_{3000}$ and mPEG$_{5000}$-PCL$_{3000}$-FA polymers to afford structurally stable nanoparticle ZNNPs and ZNNPs@FA in biological medium. Mechanistically, these nanoprobes can undergo nucleophilic substitution reaction with $H_2S$ molecules to specifically generate NIR conversion ($F_{1070} \rightarrow F_{720}$) and ratiometric photoacoustic ($PA_{680}/PA_{900}$) signals. Importantly, the ratiometric photoacoustic signal change presents a large Stokes shift ($\Delta\lambda = 220$ nm), which can greatly reduce the cross-interference between $PA_{680}$ and $PA_{900}$, allowing for sensitive and accurate detection of endogenous $H_2S$ level in acute hepatotoxicity, cerebral hemorrhage, and colorectal tumor of living mice. We also find that ZNNPs@FA could simultaneously deplete the mitochondrial $H_2S$ level in cancer cells leading to remarkable reduction of glycolysis and severe mitochondrial damage, together with the activated photodynamic effect, resulting in efficient suppression of colorectal tumor in vivo.

## Results

### Design and formulation of $H_2S$-activatable NIR-II nanoprobes.
Two $H_2S$-activatable NIR-II nanoprobes ZNNPs and ZNNPs@FA were rationally designed and prepared. We employed NIR-II fluorophore ZM1068-NB to fabricate the probes because 4-nitrobenzoic ester of ZM1068-NB can specifically react with $H_2S$ to release the enolic heptamethine cyanine that spontaneously undergoes the keto-enol tautormerization to form ZM1068-ketone product accompanied with significant absorption and emission spectral shift (Supplementary Fig. 1), showing noticeable NIR conversion ($F_{1070} \rightarrow F_{720}$) and ratiometric PA ($PA_{680}/PA_{900}$) response to $H_2S$. Moreover, the photodynamic effect of ZM1068-NB is substantially silenced at the "off" state. In the presence of $H_2S$, the leaving of 4-sulmeryl nitrobenzoate is rapidly triggered and ZM1068-ketone is simultaneously produced with distinct blue-shift absorption ($\lambda_{abs} = 650$ nm), which can activate ROS generation of ZM1068 under 660 nm laser irradiation, endowing it with excellent photodynamic effect as a promising $H_2S$-activatable photosensitizer for photodynamic therapy (PDT) application. To improve the tumor targeting capability, ZM1068-NB was encapsulated by copolymer mPEG$_{5000}$-PCL$_{3000}$ and mPEG$_{5000}$-PCL$_{3000}$-FA containing folic acid specifically targeting to folic acid receptor that are generally overexpressed in the majority of cancers to afford nanoparticles ZNNPs@FA. Briefly, ZM1068-NB was synthesized through the intermolecular esterification of 4-nitrobenzoic chloride and *meso*-hydroxyl heptamethine cyanine according to the methods previously reported with minor modification (Supplementary Fig. 2)[40,42–45], and its chemical structure was confirmed by NMR spectrometry (Supplementary Fig. 3) and mass spectrometry (MS) (Supplementary Fig. 4). To improve the biocompatibility and dispersion of nanoparticles in water, the hydrophobic ZM1068-NB was wrapped by amphiphilic mPEG$_{5000}$-PCL$_{3000}$ or mPEG$_{5000}$-PCL$_{3000}$-FA copolymers to give the desired nanoprobes with well-defined size distribution in aqueous solution. For ZNNPs, the encapsulating ratio between ZM1068-NB and mPEG$_{5000}$-PCL$_{3000}$ is optimized at 1:4 by mass. For ZNNPs@FA, the optimized ratio of ZM1068-NB, mPEG$_{5000}$-PCL$_{3000}$, and mPEG$_{5000}$-PCL$_{3000}$-FA is 1:3.8:0.2.

### Chemical characterization of ZNNPs and ZNNPs@FA.
Structures of the resulting probes were first characterized by fourier transform infrared (IR), zeta potential analysis, and UV-vis absorption. IR spectral results in Supplementary Fig. 5a show that the characteristic absorption peaks at 3200–3300 and 1637 cm$^{-1}$ corresponding to amine and ester carbonyl groups, respectively.

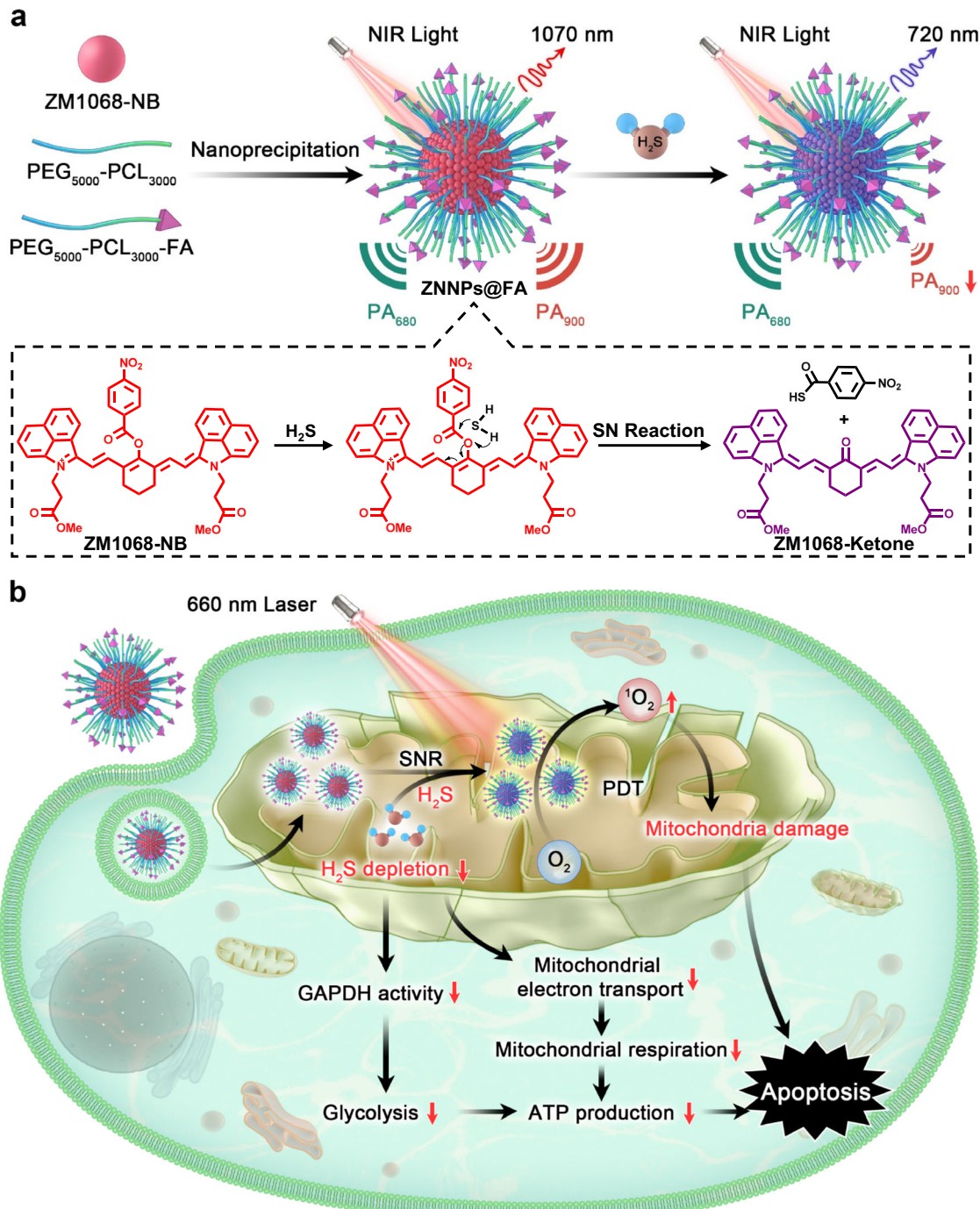

**Fig. 1 Schematic illustration of the design of nanoprobes and colorectal tumor treatment. a** The fabrication of ZNNPs@FA and principle of quantitative visualization of H₂S. **b** Consumption of endogenous H₂S in colorectal cancer inhibits ATP synthesis and activates the photodynamic effect of ZNNPs@FA.

Besides, another maximum absorption peak at 1100 nm recorded from both ZNNPs and ZNNPs@FA should correspond to ether bond, which is quite consistent with IR absorption profile of ZM1068-NB. The zeta potential analysis shows the charges of mPEG$_{5000}$-PCL$_{3000}$, ZNNPs, and ZNNPs@FA are $-21 \pm 0.63$ mV, $29.3 \pm 0.93$, and $24.6 \pm 0.6$ mV (Supplementary Fig. 5b), respectively, indicating that the loading of ZM1068-NB in polymers can yield significant positive surface change, which endows probes mitochondria-targeting ability. The morphology study indicated that both ZNNPs and ZNNPs@FA presented in aqueous solution as mono-dispersed nanoparticles with similar mean hydrodynamic size of 65 and 70 nm as determined by dynamic light scattering (DLS) (Fig. 2a and Supplementary Fig. 6a). Similarly, their average diameters are $70 \pm 3.5$ nm and $77 \pm 5$ nm according to transmission electron microscope (TEM) results in Fig. 2b and Supplementary Fig. 6b. Besides, the stability of nanoprobes was also evaluated by monitoring the hydrodynamic size change using DLS. The results shown in Supplementary Fig. 7 reveal that both nanoprobes possess excellent stability upon storage in phosphate-buffered saline (PBS, pH = 7.4) at 37 °C for 7 days.

**Fluorescence and PA responsiveness of nanoprobes towards H₂S.** We next investigated the responsiveness of probes towards

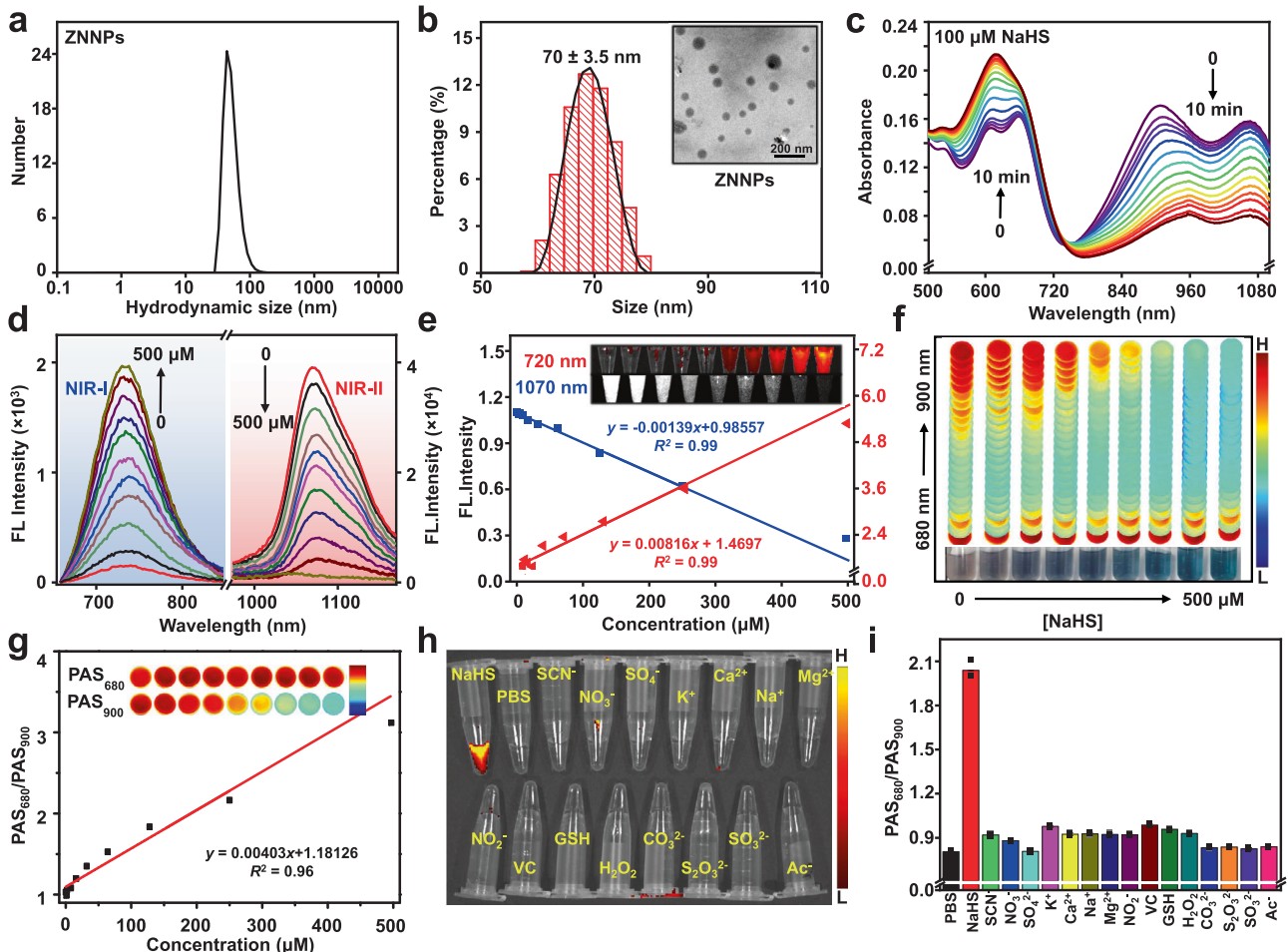

**Fig. 2 Characterization and $H_2S$ specificity of ZNNPs in PBS buffer. a** DLS data of ZNNPs in PBS buffer (pH = 7.4). **b** TEM image and size distribution of ZNNPs. **c** Normalized absorption of ZNNPs (14 μg/mL) upon incubation with NaHS (100 μM) at R.T. for 10 min. **d** Fluorescence spectra and **e** linear fitting of fluorescence intensity of ZNNPs (14 μg/mL) with various concentrations of NaHS ranging from 0 to 500 μM at R.T. for 10 min (Right axis: 1070 nm, Left axis: 720 nm). **f** Photoacoustic signal profiles and **g** linear fitting of PAS intensity of ZNNPs (14 μg/mL) with various concentrations of NaHS ranging from 0 to 500 μM at R.T. for 10 min. **h** Fluorescence images and **i** ratiometric photoacoustic signals ($PAS_{680}/PAS_{900}$) of ZNNPs (14 μg/mL) incubated with different biological species (PBS buffer, 100 μM NaHS, 1 mM $SCN^-$, 1 mM $NO_3^-$, 1 mM $SO_4^{2-}$, 1 mM $K^+$, 1 mM $Ca^{2+}$, 1 mM $Na^+$, 1 mM $Mg^{2+}$, 1 mM $NO_2^-$, 1.25 mM VC, 10 mM GSH, 1 mM $H_2O_2$, 1 mM $CO_3^{2-}$, 1 mM $S_2O_3^{2-}$, and 1 mM $SO_3^{2-}$, and 1 mM $Ac^-$) at 37 °C for 10 min. Data was presented as mean ± s.d. (n = 3 independent samples). Source data are provided as a Source Data file.

$H_2S$, ZNNPs (20 μL, 1.4 mg mL$^{-1}$) was incubated with 100 μM NaHS (1.98 mL) in PBS buffer followed by real-time absorption and fluorescence measurements. The absorption spectra in Fig. 2c indicate that ZNNPs aqueous solutions initially exhibit two major absorption bands at 680 and 900 nm. When they react with NaHS, the absorption intensity at 680 nm increases slightly, while the absorption at 900 nm decreases gradually with the increase of incubation time. If the same concentration of ZNNPs solution was treated with various amounts of NaHS ranging from 0 to 500 μM, similar absorption change of ZM1068-NB to NaHS is observed (Supplementary Fig. 8). Additionally, the reaction rate of probe ZNNPs with $HS^-$ increased significantly with the increasing amount of NaHS (Supplementary Fig. 9). Moreover, high-performance liquid chromatography (HPLC) analysis confirms that the whole reaction process is almost completed through approximately 10 min incubation accompanied by the color change from black to deep blue simultaneously (Supplementary Fig. 10). The reaction mechanism between ZM1068-NB and $HS^-$ was further verified by MS and $^1H$ NMR measurements (Supplementary Fig. 11 and Supplementary Fig. 12). As expected, the m/z peak of ZM1068-ketone is 623.33 (calculated value was 624.26), and the m/z peak of 4-sulmeryl nitrobenzoate is 182.06

(calculated value was 183.00). The fluorescence responsiveness of ZNNPs towards $H_2S$ was then evaluated in aqueous solution. Fluorescence spectra in Fig. 2d indicate that the aqueous solution of ZNNPs emits strong NIR-II fluorescence at 1070 nm initially (QY = 0.1%). However, in the presence of NaHS with increasing concentrations, the fluorescence intensity (F) of 1070 nm decreases gradually. On the contrary, significant increase of fluorescence at 720 nm was recorded (QY = 17.2%) (Supplementary Fig. 13a, 13b), which can be effectively suppressed by $ZnCl_2$ that is, a $H_2S$ scavenger, suggesting that ZNNPs has excellent fluorescence responsiveness towards $H_2S$. We speculate that this $H_2S$-responsive ratiometric fluorescence change should be ascribed to the conversion of ZM1068-NB to ZM1068-ketone via $HS^-$-ester nucleophilic substitution reaction. Next, we investigated the detecting sensitivity of ZNNPs to NaHS, As shown in Fig. 2e, the fluorescence intensity at 720 nm is positively correlated with the increasing concentrations of NaHS, whereas the fluorescence at 1070 nm decreases gradually. Clearly, the activated fluorescence at 720 nm shows a good linear correlation with the NaHS concentration ranging from 0 to 500 μM ($R^2 = 0.99$), according to which a detection limit down to 4.7 μM is derived (Supplementary Fig. 13c), implying that ZNNPs is

highly sensitive for $H_2S$ detection. In view of the aforementioned ratiometric absorption variation of ZNNPs in the presence of NaHS, we hypothesized that ZNNPs may have a specific PA responsiveness towards $H_2S$. To verify our assumption, PA signals of ZNNPs aqueous solutions containing various amounts of NaHS ranging from 0 to 500 μM were determined after 10 min incubation. The signals recorded under 680 nm (denoted as $PAS_{680}$) to 900 nm (denoted as $PAS_{900}$) excitation were compared after subtracting the signal recorded from a control solution without NaHS. As shown in Fig. 2f, Supplementary Fig. 13d and Supplementary Fig. 14, the $PAS_{680}$ remains almost unchanged, while $PAS_{900}$ is gradually decreased against the increasing concentrations of NaHS with the lowest detection limit of 0.68 μM, suggesting that $PAS_{900}$ is highly sensitive to NaHS. To reduce the interference from the surrounding condition, the ratiometric $PAS_{680}/PAS_{900}$ is plotted against the concentrations of NaHS. As shown in Fig. 2g, the $PAS_{680}/PAS_{900}$ values increases gradually with the increase of NaHS concentrations, showing a good linearity over a range from 0 to 500 μM ($R^2 = 0.96$), which may offer a useful means for quantitative detection of $H_2S$ (Supplementary Fig. 15a, 15b). To investigate the selectivity of the probes towards NaHS, ZNNPs was treated with different types of biologically relevant species under the identical condition. The results showed strong fluorescence enhancement at 720 nm was activated by NaHS accompanying the apparent decrease of absorption intensity at 900 nm but not by other species, such as $Ac^-$, $SCN^-$, $NO_3^-$, $SO_4^{2-}$, $K^+$, $Ca^{2+}$, $Na+$, $Mg^{2+}$, $NO_2^-$, VC, GSH, $H_2O_2$, $CO_3^{2-}$, $S_2O_3^{2-}$, and $SO_3^{2-}$ (Fig. 2h, Supplementary Fig. 15c and 15d). Consistently, compared with other species, PA signal of ZNNPs aqueous solution at 900 nm is dramatically suppressed upon incubation with NaHS (Supplementary Fig. 15e), and the corresponding $PAS_{680}/PAS_{900}$ ratio reaches up to 2, which is around 2.4-fold higher than others (Fig. 2i), revealing that ZNNPs exhibits ratiometric PA sensing capability for $H_2S$. Collectively, all these results firmly imply that this smart probe has a great potential to detect $H_2S$ with high sensitivity and selectivity in vitro.

**Fluorescence and PA imaging of $H_2S$ in HCT116 cells**. With the above exciting results, we intended to explore the potential of the probes for the detection of $H_2S$ in living cells. The cytotoxicity of both ZNNPs and ZNNPs@FA to human embryonic kidney 293 (HEK293) cells was first evaluated through the widely used methyl thiazolyl tetrazolium (MTT) assay. The overall cell viability of HEK293 cells incubated with even 100 μg mL$^{-1}$ of ZNNPs or ZNNPs@FA for 24 h still remained above 85% (Supplementary Fig. 16), indicative of negligible cytotoxicity to HEK293 cells. Moreover, no significant red cell damage was determined by the hemolysis assays for both probes even at the concentration of 160 μg mL$^{-1}$ (Supplementary Fig. 17), suggesting they essentially have excellent biocompatibility. In the following experiments, we chosen HCT116 as $H_2S$-rich cancer cells. The confocal laser scanning microscopy (CLSM) imaging was conducted for HCT116 living cells after being incubated with ZNNPs for 4 h. As shown in Fig. 3a, obvious red fluorescence was determined in HCT116 cells. To verify the afforded fluorescence was indeed activated by endogenous $H_2S$, HCT116 cells were pre-incubated with $ZnCl_2$ followed by CLSM imaging. The fluorescence signals were substantially suppressed in comparison with the control cells. Moreover, if the cells were pre-incubated with extraneous NaHS (1 mM) followed by ZNNPs treatment, remarkable fluorescence signals up to 4.6-fold of the control cells was recorded. Likewise, the cells pre-treated with L-Cysteine (L-Cys), a precursor for the biosynthesis of $H_2S$, to stimulate the expression of intracellular $H_2S$ showed significantly higher red

fluorescence than control cells, which could be effectively inhibited when adding DL-propargyl glycine (PAG, 50 μg mL$^{-1}$) to suppress the activity of endogenous cystathionine-g-lyase (CSE)[46], This can be further confirmed by the $F_{1070}$ imaging ($Ex = 808$ nm, $Em = 1070$ nm) results shown in Supplementary Fig. 18. Together, these results highly suggest that the fluorescence enhancement is specifically activated by $H_2S$. Since $H_2S$ is usually over-expressed in inflammation disease[12,13], as expected, the lipopolysaccharide (LPS)/L-Cys-treated HCT116 cells showed intensive fluorescence (2.9-fold of the control cells), which can be nearly suppressed by PAG (50 μg mL$^{-1}$) completely, implying that LPS is able to positively up-regulate the CSE expression and $H_2S$ production in living cells (Fig. 3b). To further evaluate the detection capability of ZNNPs to HCT116 cells through PA imaging, the cellular pellets from above seven groups were collected and lysed to aqueous solutions for PA signal measurement. As illuminated in Fig. 3c, all samples exhibited strong PA signals under excitation of 680 nm ($PAS_{680}$), whereas the NaHS-pretreated cells incubated with ZNNPs showed the lowest PA signal under the illumination of 900 nm ($PAS_{900}$) compared to other groups, demonstrating that $PAS_{900}$ is more sensitive to $H_2S$. Furthermore, the quantitative ratio of $PAS_{680}/PAS_{900}$ for the cells receiving NaHS was calculated to be 1.6-fold of the control sample (Fig. 3d). Collectively, all these results strongly demonstrate that ZNNPs probe is highly useful for specific detection of endogenous $H_2S$ in living cells through NIR and PA imaging, which encourages us to further assess its potential for in vivo imaging of $H_2S$.

**Fluorescence imaging of hepatic $H_2S$ in mice liver**. Inspired by the above exciting results, we next evaluated the performance of our probe for in vivo visualization of endogenous $H_2S$. Since $H_2S$ is active in the liver as a signaling agent, and closely associates to various liver diseases[47–49]. Hence, real-time monitoring of hepatic $H_2S$ level would be greatly meaningful for early diagnosis and elucidation of liver diseases. We next attempted to monitor the endogenous $H_2S$ in living mice using ZNNPs (Fig. 4a). The mice were intraperitoneally (i.p.) administrated with PBS first and then intravenously (i.v.) injected with ZNNPs (10 mg/kg) through the tail vain after 30 min followed by systemic fluorescence scanning with IVIS imaging system. Figure 4b shows series of representative fluorescence images of mice acquired at selected time intervals after injection of ZNNPs. Apparent fluorescence signals at both 720 nm and 1070 nm were noted in the liver region, and the fluorescence intensity gradually increased over time, indicating that large amounts of probes accumulate in the liver of mice. Intriguingly, NIR-II fluorescence of the probes should theoretically become weaker encountering $H_2S$, while the NIR-II images at 1070 nm are significantly superior to the ones at 720 nm, which should be attributed to the deeper tissue penetration capability of NIR-II fluorescence, firmly implying that NIR-II imaging is more reliable and sensitive than visible NIR-I imaging extremely in deep tissue bioimaging application. To further validate the fluorescence signals were in deed activated by $H_2S$ in liver, mice were i.p. administrated with L-Cys (100 μL, 6 mM) to upregulate the production of hepatic $H_2S$ first followed by injection of ZNNPs. The fluorescence signals at 720 nm ($F_{720}$) became significantly brighter than the mice with PBS, and the signal-to-background ratio (SBR) of livers increased 5-fold at 1.5 h post-injection, while the fluorescence at 1070 nm ($F_{1070}$) dramatically decreased (Fig. 4b). Moreover, when the mice were i.p. injected with PAG (100 μL, 2 mg/mL) to suppress the production of $H_2S$ through inhibiting CSE activity. Both $F_{720}$ and $SBR_{720}$ of the liver reduced significantly. However, the fluorescence and SBR at 1070 nm enhanced remarkably with the increase of time (Fig. 4c).

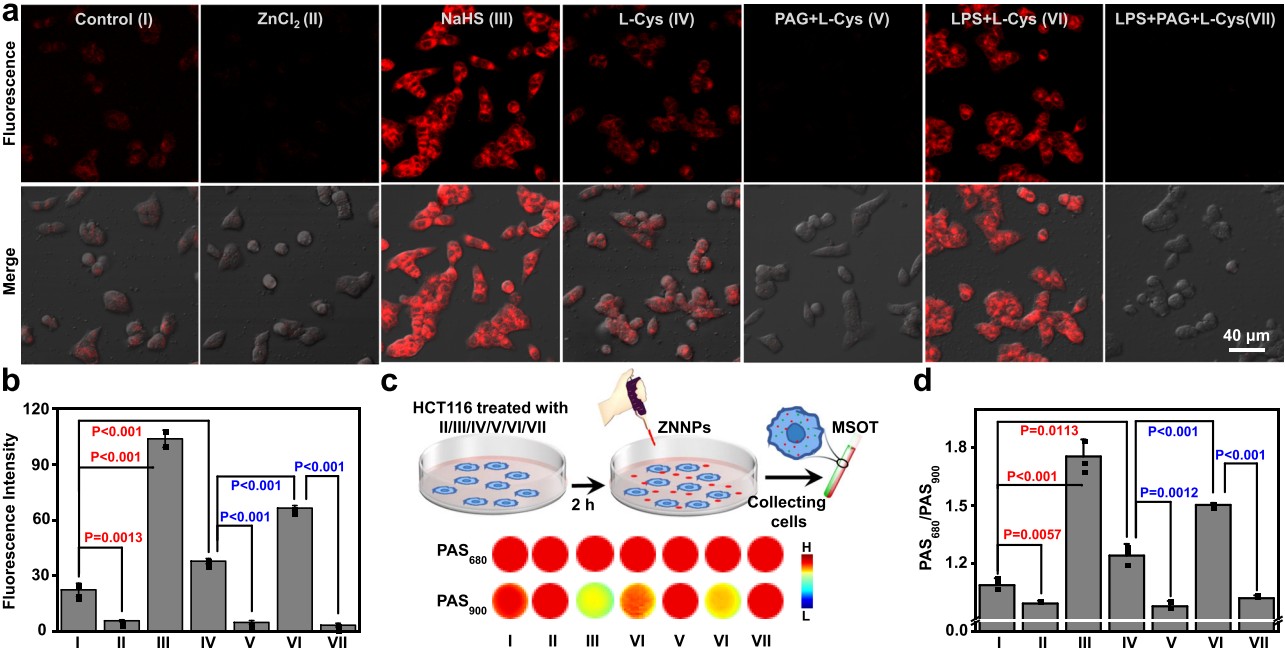

**Fig. 3 Visualization of H₂S in HCT116 cells. a** Confocal fluorescence images and **b** quantitative fluorescence intensity of HCT116 cells. **c** Photoacoustic images and **d** ratiometric photoacoustic signals (PAS₆₈₀/PAS₉₀₀) of H₂S in HCT116 cells. Cells were treated with PBS buffer, ZnCl₂ (40 μg/mL, 10 min), NaHS (1 mM, 1 h), L-Cys (24 μg/mL, 1 h), PAG (50 μg/mL, 0.5 h) + L-Cys (24 μg/mL, 1 h), LPS (1 μg/mL, 6 h) + L-Cys (24 μg/mL, 1 h), or PAG (50 μg/mL, 0.5 h) + LPS (1 μg/mL, 6 h) + L-Cys (24 μg/mL, 1 h). Data are presented as mean ± s.d. (*n* = 3). Statistical differences were analyzed by Student's two-sided t-test. Source data are provided as a Source Data file.

These imaging results were further verified by the ex vivo imaging of the resected organs including heart, liver, spleen, lung, and kidney (Supplementary Fig. 19), again revealing that high level of H₂S presents in the liver of mice. Collectively, all above evidences strongly demonstrate that ZNNPs is useful for sensitive detection of hepatic H₂S level in living mice.

**PA imaging of hepatic H₂S in mice liver.** Although fluorescence imaging has a great potential in bioimaging, the shallow tissue penetration seriously limits its clinical translation. In contrast, PA imaging synergistically integrated fluorescence and ultrasound imaging together shows excellent deep tissue imaging capability with high spatial resolution. To evaluate the PA responsiveness of the probe towards hepatic H₂S in living mice, above three groups of mice treated with different chemical species (PBS, L-Cys, and PAG) were further imaged upon illumination at wavelengths of 680 nm and 900 nm. As shown in Fig. 4d and Supplementary Fig. 20, both PA₆₈₀ and PA₉₀₀ signals of livers in three groups increased gradually with the prolongation of time and reached the maximum at around 1.5 h post-injection of ZNNPs due to the increasing accumulation of ZNNPs in the livers. In particular, the mice with L-Cys showed the highest PA₆₈₀ signal, while the lowest PA₆₈₀ signals were detected from the mice with PAG. On the contrary, the strongest PA₉₀₀ signals were determined from the mice receiving PAG and the lowest PA₉₀₀ from the mice of L-Cys, suggesting that the variation of PA at 680 nm and 900 nm can accurately reflect the expression level of H₂S in mouse liver. Moreover, the ratiometric PA (PA₆₈₀/PA₉₀₀) of the liver receiving L-Cys enhanced most significantly and remained constant from 1 to 3 h (Fig. 4e), which is approximately 2.8-fold of the livers with PBS, suggesting that the PA₆₈₀/PA₉₀₀ of ZNNPs at 1.5 h post-injection is suitable for quantifying the concentration of H₂S in livers. Collectively, these results demonstrated that probe ZNNPs can be potentially applied for in vivo quantitatively imaging of endogenous H₂S through ratiometric PAI.

**Ratiometric PA quantification of hepatic H₂S in drug-induced damaged liver.** In light of above exciting findings, the upregulation of hepatic H₂S in drug-treated livers were further monitored by ratiometric PAI. The mice were i.p. injected with different amounts of L-Cys followed by i.v. administration of ZNNPs (10 mg/kg) via the tail vain after 30 min. After 1.5 h, both fluorescence and PA images were collected under the corresponding excitation wavelengths. As expected, both F₇₂₀ and PA₆₈₀ signals in mice livers enhanced gradually with the increase dosages of L-Cys, whereas the F₁₀₇₀ and PA₉₀₀ signals remarkably decreased (Fig. 5a, b and Supplementary Fig. 21a), revealing that L-Cys can effectively upregulate the production of hepatic H₂S. As the photoacoustic imaging holds great advantages particularly in deep tissue penetration, we next systematically evaluated the relationship of the PA signal to H₂S level. As shown in Fig. 5c, the ratiometric PA₆₈₀/PA₉₀₀ increases positively against the increasing dosages of L-Cys in a range of 0 to 5 mM (100 μL). For the livers with 5 mM L-Cys treatment, the PA₆₈₀/PA₉₀₀ signal was around 2.25-fold higher than that of normal mice liver. In addition, we also determined the concentrations of H₂S in the livers with various doses of L-Cys using the commercial kit. Figure 5d indicates that the production of H₂S in livers increased gradually with the increasing dosage of L-Cys used, again confirming that L-Cys can promote the production of endogenous H₂S. More notably, a linear relationship ($R^2 = 0.98$) between ratiometric PA₆₈₀/PA₉₀₀ and concentrations of endogenous H₂S was derived, as given in Fig. 5e. This relationship may allow us to quantitatively correlate the integrated ratiometric PA signals with the concentrations of H₂S. Next, to further evaluate whether above quantitative method can be used to assess the degree of drug-induced liver damage, the mice were i.p. administrated with different dosages of metformin ranging from 0 to 5 mg/day for seven days followed by i.v. injection of ZNNPs (10 mg/kg) after 30 min. Similar to aforementioned L-Cys-treated mice, the same trend of fluorescence and PA variation was determined (Supplementary Fig. 21b). Figure 5f, g show that the metformin

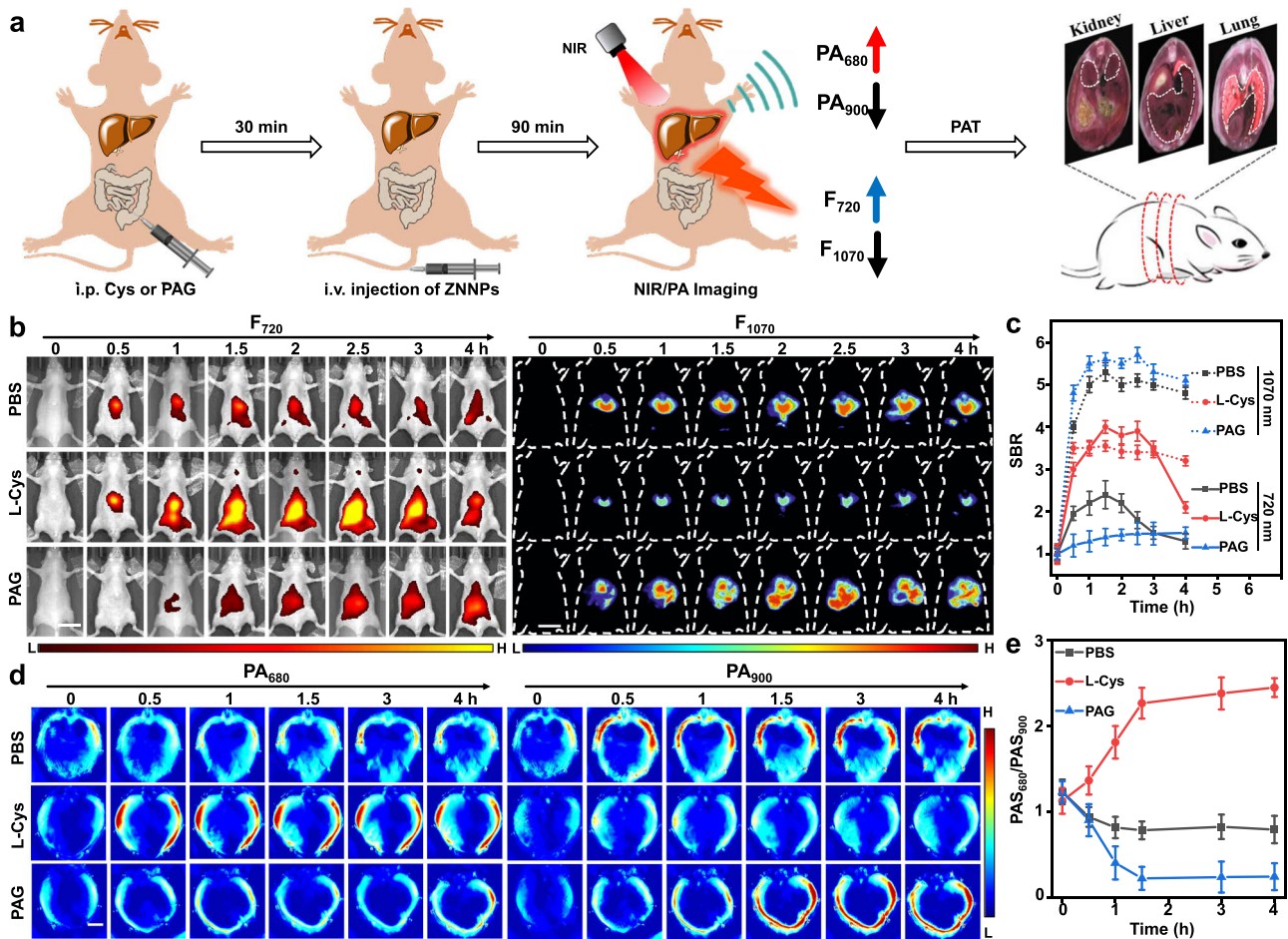

**Fig. 4 In vivo fluorescence and photoacoustic imaging of H₂S in mouse liver.** Balbc-Nu mice were intraperitoneally injected with saline (100 µL), L-Cys (6 mM, 100 µL) or PAG (2 mg/mL, 100 µL) to construct the mice with different levels of H₂S in livers. **a** Schematic description for regulation and imaging of hydrogen sulfide concentration in mouse liver by photoacoustic tomography (PAT). **b** Fluorescence images and **c** average fluorescence intensity at 720 and 1070 nm in region of liver with ZNNPs (10 mg/kg, i.v.) injection ($n = 5$). **d** PA images and **e** ratio photoacoustic signal (PAS₆₈₀/PAS₉₀₀) in region of liver treated with ZNNPs (10 mg/kg, i.v.). The scale bars in fluorescence and PA images were 2 cm and 0.5 cm, respectively. Data are presented as mean ± s.d. ($n = 5$). Source data are provided as a Source Data file.

treatment could cause significant enhancement of PA₆₈₀ and decrease of PA₉₀₀ in mouse livers. Accordingly, the PA₆₈₀/PA₉₀₀ in the livers of mice remarkably enhanced with the increasing doses of metformin. For example, 5 mg/day dosage of metformin could lead to ~2.2-fold PA₆₈₀/PA₉₀₀ signal enhancement compared to the control mice without drug treatment, suggesting that metformin can upregulate the production of H₂S in mouse liver. Moreover, to verify whether the H₂S level in mouse livers can be speculated according to the afore-derived linear relationship between PA₆₈₀/PA₉₀₀ and H₂S concentrations, we input the measured PA₆₈₀/PA₉₀₀ values in Fig. 5g into above formula to afford the corresponding concentrations of H₂S, and further correlate them with the drug doses to give a new relationship shown in Fig. 5h. To further validate the accuracy of the speculated H₂S concentration, the mouse livers with different treatments were dissected and lysed for H₂S content detection with the commercial kit. As expected, the measured concentrations of H₂S are quite close to the calculated values. These evidences firmly prove the feasibility and accuracy of this ratiometric PA-based imaging approach for quantitative detection of endogenous H₂S in vivo.

**NIR/PA dual-modality imaging of H₂S in the injured brain of mice.** It is well known that H₂S is one of the important signaling

molecules and plays a key role in the central nervous system[4,50]. However, the accurate detection of endogenous H₂S level in brain still remains a huge challenge. We further evaluated the capability of probe ZNNPs to detect the trace amount of H₂S in the injured brain of mice. The brain-injured mice were i.v. injected with ZNNPs (10 mg/kg) through tail vain. After 4 h post-injection, the mouse brain was imaged by both fluorescence and PA imaging systems at several time internals. The intensity of F₇₂₀ signals in the brain were remarkably increased over time, while the F₁₀₇₀ signals remained weak all the time (Supplementary Fig. 22), which is quite consistent with the fluorescence imaging results of mouse livers in Fig. 4c. Similarly, the intensity of PAS₆₈₀ signals was significantly increased against time after intravenous injection of the probes and reached a plateau 4 h post-injection, whereas only weak PAS₉₀₀ signal was recorded over time (Supplementary Fig. 23a). The ratiometric PA₆₈₀/PA₉₀₀ of the mouse brain enhanced gradually with the increase of time and reached a maximum with 2.4 ratio at 4 h post-injection. These results highly imply that the H₂S level of the injured region in the mouse brain is significantly upregulated. Moreover, the PA₆₈₀/PA₉₀₀ value of 4 h post-injection was imported into the afore-derived relationship equation to quantify H₂S level in the injured brain region. The concentration of endogenous H₂S is approximately 5.29 µmol/g protein. To further verify the signals come from the

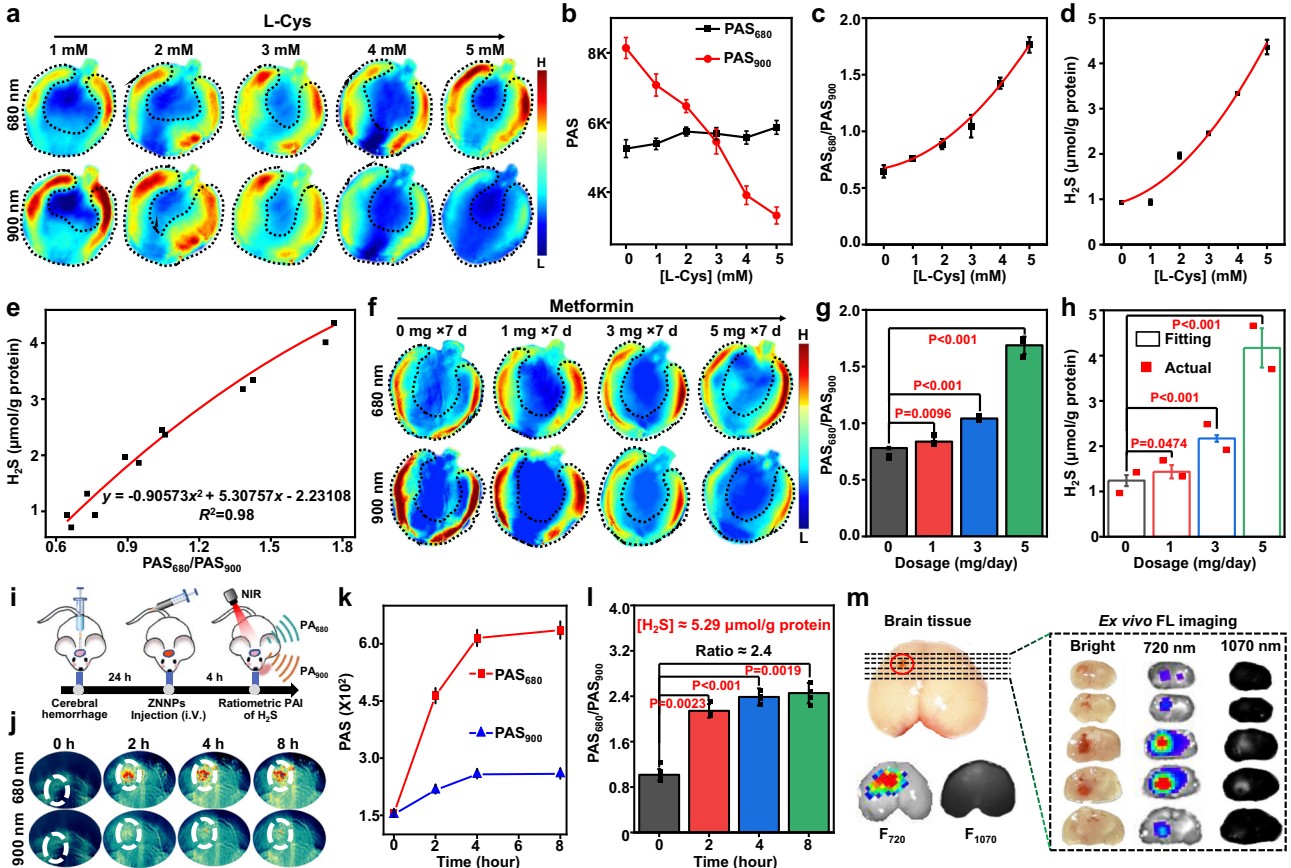

**Fig. 5 Quantitative imaging of endogenous H₂S in vivo. a** PA images and **b** PA signals intensity of mouse livers at 680 and 900 nm, **c** Ratiometric PA signal PAS$_{680}$/PAS$_{900}$ against various treatments (i.p.) of L-Cys (0, 1, 2, 3, 4, and 5 mM, 100 μL) and i.v. injection of ZNNPs (10 mg/kg) after 30 min. Data are presented as mean ± s.d. ($n = 5$ independent mice). **d** Actual H₂S concentration in mouse liver in **a**. **e** Derived equation for endogenous H₂S quantification. **f** PA images, **g** Ratiometric PAS$_{680}$/PAS$_{900}$ signals and **h** H₂S concentrations (fitting & actual) against various treatments (i.p.) of metformin (0, 1, 3, 5 mg in 100 μL PBS buffer) for 7 days and i.v. injection of ZNNPs (10 mg/kg). Data are presented as mean ± s.d. ($n = 4$ independent mice), data of "Actual" are presented as actual values ($n = 2$ independent mice). **i** Schematic description of ICH model construction and detection of endogenous H₂S. **j** PA images, **k** PA intensity at 680 and 900 nm and **l** ratiometric PAS$_{680}$/PAS$_{900}$ signals of ICR mice with i.v. injection of ZNNPs (10 mg/kg) ($n = 3$). **m** Fluorescence images of mice brain slice in **j**. Statistical differences were analyzed by Student's two-sided t-test. Source data are provided as a Source Data file.

injured brain tissues, we subsequently conducted ex vivo fluorescence imaging of the mouse brain resected from the mice. Figure 5m shows that the injured tissues in the resected brain can be obviously identified by naked eyes, and exhibit bright fluorescence signals both at 720 nm and 1070 nm (Supplementary Fig. 23b and 23c). Furthermore, both HE and Nissl staining results clearly indicate that significant brain tissue damage presented in the mice with intracerebral hemorrhage (ICH) (Supplementary Fig. 23d and 23e), confirming that both fluorescence and PA signals we detected in vivo are essentially derived from the injured brain tissues. Collectively, these exciting results strongly demonstrate that probe ZNNPs has a great potential for sensitive detection of the injured brain in mice.

**ZNNPs@FA depletes intracellular H₂S and suppress the proliferation and migration of HCT116 cells.** Numerous researches have demonstrated that H₂S can regulate the angiogenesis and cell proliferation, and is over-expressed in colorectal cancers[1,24,51]. To investigate whether the depletion of endogenous H₂S can induce cell death, we first evaluated the level of H₂S in HCT116 colorectal cancer cell. The equal amount of ZNNPs@FA (20 μg/mL) was incubated with HCT116 and HEK293 cells, respectively, for 4 h followed by CLSM imaging. HCT116 cells

showed obviously higher red fluorescence compared to HEK293 cells (Supplementary Fig. 24a and 24b), suggesting that HCT116 cancer cells have higher H₂S production than normal cells. In light of the results we obtained previously, ZNNPs nanoprobe could not only be activated by H₂S resulting in ratiometric fluorescence and PA signals, but also simultaneously consumes the intracellular H₂S that plays an important role in promoting the growth and proliferation of colorectal cancer and has been considered as a tumor growth factor and anticancer drug target (Supplementary Fig. 24c)[15,51–56]. Hence, we hypothesize that the depletion of intracellular H₂S may have an influence on the cell proliferation. To this end, we next assessed the effect of the deletion of endogenous H₂S by ZNNPs@FA on cancer cell growth and migration by real-time cell analyzer (xCELLigence) and scarification test. As shown in Fig. 6a, in comparison with two control groups of HCT116 cells, namely control and ZNNPs@FA + NaHS, the growth and proliferation of HCT116 cells receiving ZNNPs@FA were significantly inhibited (Supplementary Fig. 25c). Meanwhile, the migration capability of HCT116 cells treated with ZNNPs@FA or ZnCl₂ was clearly impaired in comparison with the control cells without any treatment (Supplementary Fig. 25a, 25b). To confirm that ZNNPs@FA enables indeed depletion of H₂S again, different amounts of ZNNPs@FA ranging from 0 to 100 μg mL⁻¹ were

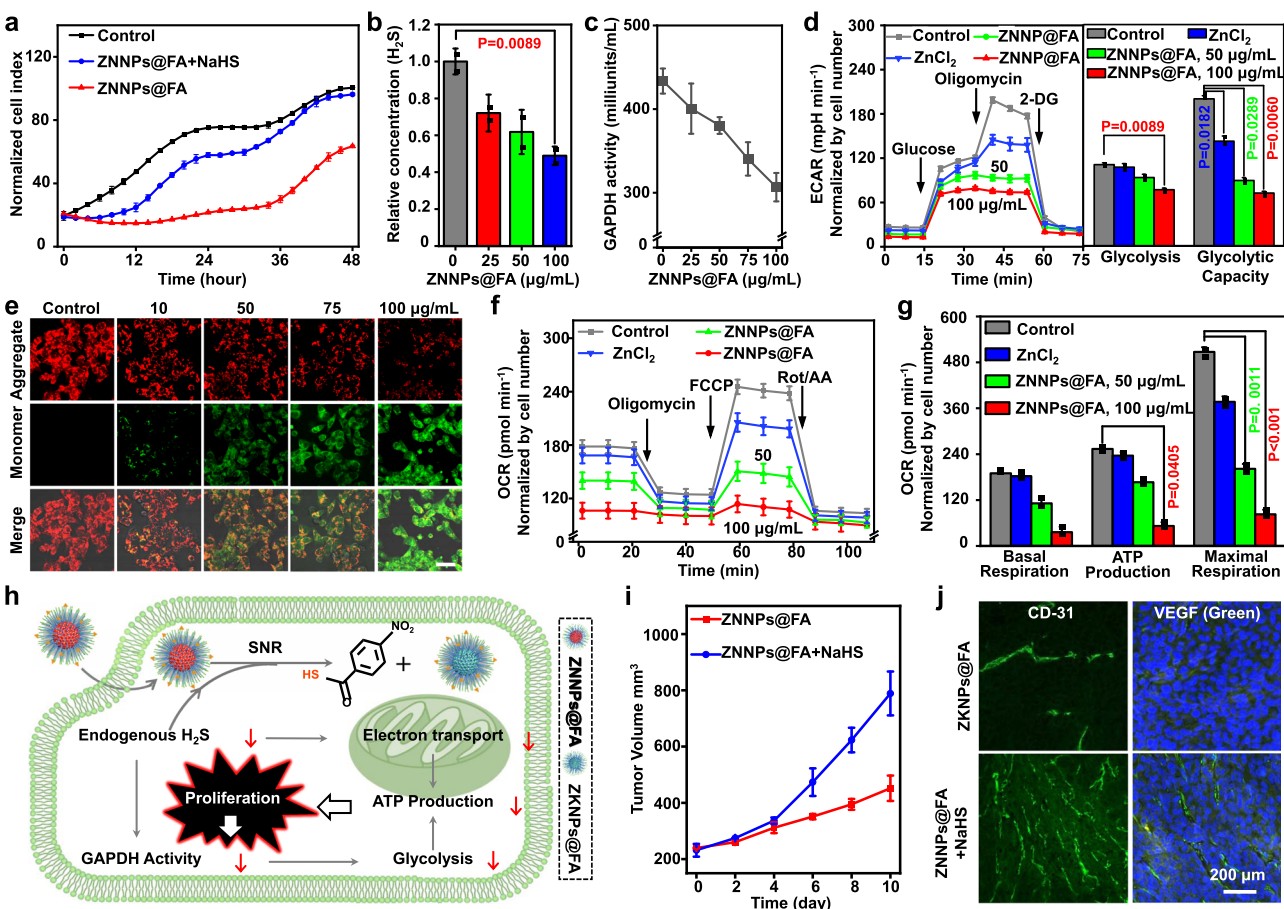

**Fig. 6 ZNNPs@FA induces HCT116 cell death via intracellular H$_2$S depletion. a** Proliferation assays of HCT116 cells with different treatments (20 μg/mL ZNNPs@FA or ZNNPs@FA + 100 μM NaHS) for 4 h. **b** H$_2$S concentration determination of HCT116 cells treated with different concentrations (0, 25, 50, 100 μg/mL) of ZNNPs@FA for 24 h. **c** GAPDH activity in various groups of cells. **d** ECAR quantification of cells with various treatments (50, 100 μg/mL ZNNPs@FA, or 20 μg/mL ZnCl$_2$ for 6 h). **e** Mitochondrial membrane potential of HCT116 cells after 24 h of incubation with different concentrations (0, 10, 50, 75, and 100 μg/mL) of ZNNPs@FA measured by JC-1 staining. Scale bar = 50 μm. **f** Cell mitochondrial stress and **g** OCR determination of cells treated with various treatments (50, 100 μg/mL ZNNPs@FA, or 20 μg/mL ZnCl$_2$) for 6 h. **h** The mechanism on H$_2$S depletion inhibiting the proliferation of HCT116 cells. **i** Tumor growth profiles. **j** Immunofluorescence staining of tumorous tissues with antibodies against CD-31 and VEGF. Data are presented as mean ± s.d. ($n = 3$). Statistical differences were analyzed by Student's two-sided t-test. Source data are provided as a Source Data file.

incubated with HCT116 cells for 24 h followed by measuring intracellular H$_2$S concentration with commercial kit. Compared with the control cells, the addition of ZNNPs@FA could significantly suppress the production of intracellular H$_2$S in a concentration-dependent manner. When the cells were treated with 100 μg mL$^{-1}$ of ZNNPs@FA, the intracellular H$_2$S level decreased to 52% of the initial one (Fig. 6b), revealing that ZNNPs@FA can efficiently inhibit the migration and proliferation of HCT116 cells through depleting the endogenous H$_2$S.

**Depletion of endogenous H$_2$S by ZNNPs@FA alters glycolytic metabolism of HCT116 cells.** Since the downregulation of endogenous H$_2$S can affect the glycolysis of cells leading to suppression of GAPDH activity[45,47,57], in order to dissect whether the depletion of endogenous H$_2$S by ZNNPs@FA regulates the glycolysis of HCT116 cells, we next quantified the GAPDH activity in the cells with the treatment of various concentrations of ZNNPs@FA. Surprisingly, the activity of GAPDH was significantly inhibited in a ZNNPs@FA concentration-dependent manner. Around 30% of the initial activity was blocked for the cells with ZNNPs@FA (100 μg mL$^{-1}$) (Fig. 6c), confirming that the depletion of intracellular H$_2$S enables reduction of GAPDH activity. Next, the extracellular acidification rate (ECAR) that

reflects the overall glycolytic flux was measured after the cells were incubated with ZNNPs@FA (50 μg/mL and 100 μg/mL) or ZnCl$_2$ (20 μg/mL) for 6 h. As shown in Fig. 6d, distinct reduction of ECAR was observed in the cells treated with either ZNNPs@FA or ZnCl$_2$ in a dosage-dependent manner, indicating that the glycolysis and glycolytic capacity of HCT116 cells can be interfered by ZNNPs@FA. Due to the positive charge characteristics of ZNNPs@FA on the surface, we further determined the subcellular localization of ZNNPs@FA in cells. The subcellular colocalization results indicated that the probes mainly accumulated in mitochondria (Fig. 6e and Supplementary Fig. 25d). To investigate whether ZNNPs@FA disrupts the mitochondria function and metabolic pathway, the integrity of cellular mitochondria was first determined by using JC-1 dye. As shown in Fig. 6f, the nontreated HCT116 cells showed only strong red fluorescence (JC-1 aggregates) in mitochondria, indicating that the mitochondria remains good integrity. However, weak red fluorescence and strong green fluorescence (JC-1 monomers) were obviously detected in the cells receiving ZNNPs@FA with various concentrations, implying that the mitochondrial membrane has been disrupted. Moreover, the oxygen consumption rate (OCR), an indicator of mitochondrial oxidative respiration, was measured for HCT116 cells with various treatments. In comparison with the nontreated control cells, significant OCR inhibition was observed

for the cells treated with ZNNPs@FA or ZnCl$_2$ (Fig. 6g), which is largely dependent on the concentration of ZNNPs@FA used. Additionally, the cells had a negligible responsiveness to the subsequent addition of oligomycin, an ATP synthase inhibitor, and carbonyl cyanide 4-(trifluoromethoxy) phenylhydrazone (FCCP) that can damage the proton gradient, indicating that ZNNPs@FA can reduce basal cellular respiration, ATP production, and spare respiratory capacity (Fig. 6g). Collectively, all these evidences strongly demonstrate that the ZNNPs@FA-enabled H$_2$S depletion can affect the mitochondrial function and glycolysis leading to in consequence severe suppression of HCT116 cell proliferation (Fig. 6h).

**In vivo imaging of H$_2$S depletion and suppression of HCT116 tumors**. Since H$_2$S has be found to be closely associated with colorectal tumor, to visualize the H$_2$S-depleting behavior in vivo, ZNNPs@FA (10 mg/kg) was intravenously injected into HCT116 tumor-bearing female nude mice followed by real-time fluorescence and PA imaging. The depletion of H$_2$S depicted by the fluorescence signals at 720 nm showed a remarkable increase over time and reached the maximum at 10 h post-injection (Supplementary Fig. 26a), which should be ascribed to the gradual accumulation of ZNNPs@FA as well as the depletion of endogenous H$_2$S. However, the NIR-II fluorescence at 1070 nm showed an increasing trend, but the overall enhancement was modest, which is consistent with the previous observation that H$_2$S can quench the fluorescence signals of the probes in NIR-II region. Meanwhile, the PA images of the tumors under the excitation of 680 nm and 900 nm were also acquired at several time points. Similarly, the PAS$_{680}$ signals gradually enhanced with the increasing time, whereas the increase of PAS$_{900}$ signals was very little all the time. Together, these results demonstrate that the depletion of H$_2$S in tumors can be visualized by ZNNPs@FA through both fluorescence and PA imaging. To further assess if the ZNNPs@FA-based H$_2$S depletion led to suppression of tumor growth in living mice, ZNNPs@FA (10 mg/kg) and the mixture of ZNNPs@FA and NaHS as control group were intratumorally injected into the mice bearing two HCT116 tumors on both the left and right back, respectively. The tumor growth was subsequently monitored by measuring the average tumor sizes of two groups over 10 days. As shown in Fig. 6i, j, the tumors in the control group exhibited a rapid growth over time. By huge contrast, the tumors receiving ZNNPs@FA were significantly reduced in size with around 50% tumor suppression on day 10 compared to the control group (Supplementary Fig. 27a). Furthermore, both groups of tumor tissues were resected on 10 days and sliced for immunostaining with antibodies against CD31 and VEGF. Obvious high expression of CD31 and VEGF were detected from the tumor sections of control mice. However, the tumors with the treatment of ZNNPs@FA displayed significant reduction of CD31 and VEGF expression (Supplementary Fig. 27), revealing that the depletion of H$_2$S by ZNNPs@FA truly causes effective inhibition of HCT116 tumor growth.

**Photodynamic effect of H$_2$S-activatable. ZNNPs@FA**. Encouraged by the above results, we further investigated the photodynamic effect of ZNNPs@FA with a singlet oxygen ($^1$O$_2$) sensor DPBF in vitro condition. We first determined the absorption variation of DPBF in the presence of ZNNPs@FA with or without NaHS under the laser irradiation (660 nm, 50 mW/cm$^2$) over time. The absorption intensities at 415 nm slightly decreased for the assays without NaHS, indicating little $^1$O$_2$ production. However, in the presence of NaHS, the absorption intensities of assays were dramatically decreased with the increasing concentrations of NaHS (Supplementary Fig. 28). Next, we further

measured the singlet oxygen ($^1$O$_2$) quantum yield of ZNNPs@FA in the presence of different concentrations of NaHS ranging from 0 to 300 μM. The results in Supplementary Fig. 29 demonstrate that H$_2$S can activate the $^1$O$_2$ generation of ZNNPs@FA upon laser irradiation in a dose-dependent manner. Next, we further evaluated the photodynamic effect of ZNNPs@FA in living HCT116 cells. The intracellular $^1$O$_2$ levels of several groups of cells with different treatments were monitored with 2,7-dichlorodihydrofluorescein diacetate (DCFH-DA), a non-fluorescent molecule that can diffuse into the cells and then be oxidized by $^1$O$_2$ to generate green fluorescence emissive DCF. As shown in Fig. 7a, after treated with ZNNPs@FA and 660 nm irradiation, HCT116 cells displayed obviously stronger green fluorescence than HEK293 cells (Supplementary Fig. 30a), again confirming that the endogenous H$_2$S level in HCT116 cells is higher than normal cells. Additionally, higher DCF green fluorescence signals were recorded for HCT116 cells with the treatment of exogenous NaHS or L-Cys, which could be effectively suppressed by ZnCl$_2$ (Fig. 7b), well demonstrating that $^1$O$_2$ generation capability of the current probe is specifically activated by H$_2$S. Furthermore, the integrity of cellular mitochondria was detected using JC-1 dye staining. As shown in Fig. 7c, the HCT116 cells themselves showed intensive red fluorescence (JC-1 aggregates) in mitochondria, while the cells receiving ZNNPs@FA displayed clear green fluorescence (JC-1 monomers) apart from red fluorescence (Supplementary Fig. 30b), well matching with afore-mentioned mitochondrial disruption of ZNNPs@FA to the cancer cell, which was further confirmed by TEM images. Along with the disruption of cellular mitochondria, MTT assays showed significant cell death was determined in HCT116 cells under the treatment of ZNNPs@FA and 660 nm laser irradiation (Fig. 7d). Collectively, these evidences strongly support that probe ZNNPs@FA has an excellent H$_2$S-activated photodynamic effect under light illumination to kill colorectal cancer cells.

**In vivo H$_2$S-activated PDT of colorectal tumors using ZNNPs@FA**. Inspired by the superior H$_2$S-activated photodynamic therapy capability of ZNNPs@FA in living cancer cells, we further evaluated the antitumor efficacy of combined H$_2$S depletion and PDT in HCT116 tumor-bearing BALB/c mice. The mice were randomly divided into four groups ($n = 5$), and were i.v. administrated to different combinations of treatments in which tumor receiving PBS (denoted as control), tumor receiving PBS followed by 660 nm irradiation (denoted as control +660 nm), and tumor with ZNNPs@FA (denoted as ZNNPs@FA) were set as control groups, for showing the antitumor efficacy of the experimental group ZNNPs@FA followed by 660 nm irradiation (denoted as ZNNPs@FA + 660 nm) (Fig. 8a). The tumor growth was subsequently real-time monitored by measuring the average tumor sizes of different groups over 20 days. As shown in Figs. 8b, c, rapid tumor growth was observed for the groups of control and control+660 nm in a similar trend with a rather weak tumor suppression. In contrast, the mice with ZNNPs@FA showed remarkable tumor growth inhibition, implying that the depletion of endogenous H$_2$S in tumors can suppress the tumor growth (Supplementary Fig. 31), which is consistent with the results obtained from the intratumoral injection experiments. More notably, the tumors receiving ZNNPs@FA and 660 nm irradiation were significantly reduced in size with approximately 89.3% tumor inhibition on day 20 compared to the control group (Fig. 8d). The survival rates of mice receiving different treatments indicate that ZNNPs@FA + 660 nm group exhibits greatly improved survival rate without single death over 20 days in comparison with control groups (Supplementary Fig. 32). Besides, no obvious body weight change

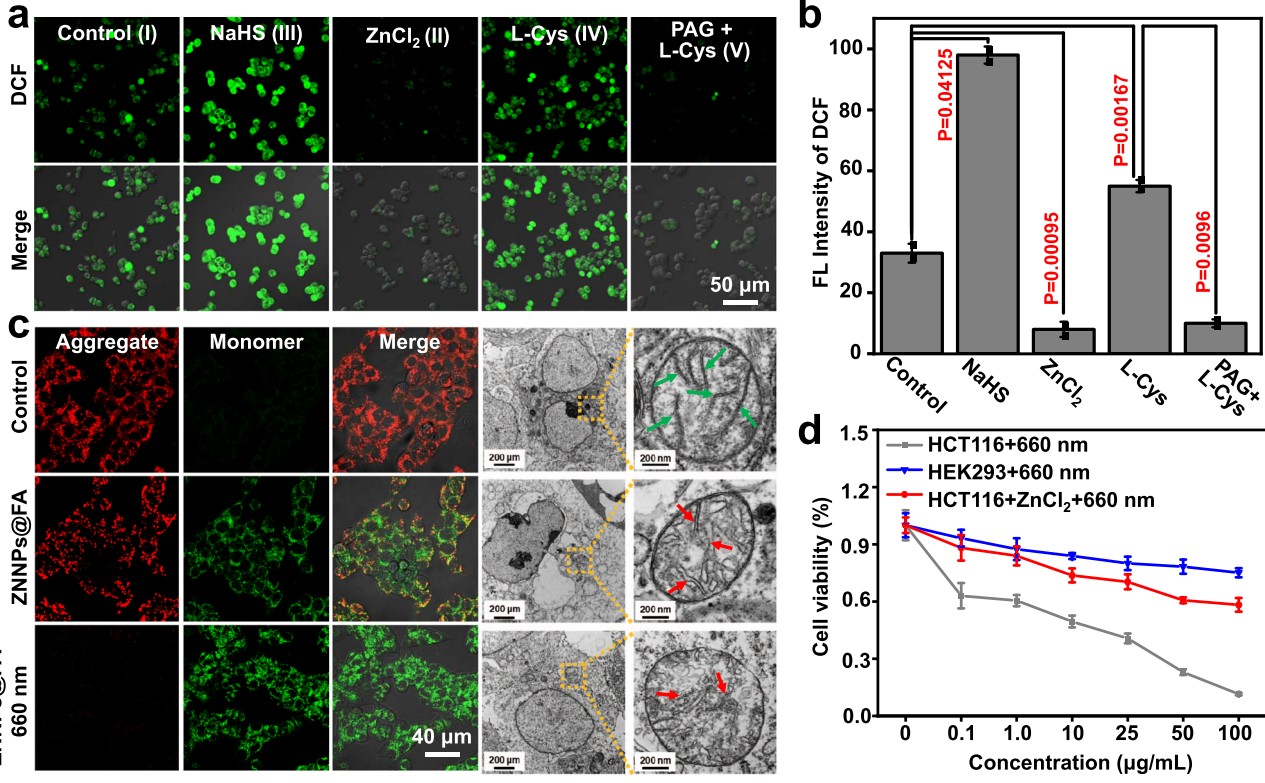

**Fig. 7 H2S-activated PDT effect of ZNNPs@FA in HCT116 cells. a** Confocal fluorescence imaging and **b** quantification of H2S-activated ZNNPs@FA enabling $^1O_2$ generation in HCT116 cells pretreated with PBS buffer, ZnCl2 (40 μg/mL, 10 min), NaHS (1 mM, 1 h), L-Cys (24 μg/mL, 1 h), PAG (50 μg/mL, 0.5 h)+L-Cys (24 μg/mL, 1 h), LPS (1 μg/mL, 6 h)+L-Cys (24 μg/mL, 1 h), or PAG (50 μg/mL, 0.5 h)+LPS (1 μg/mL, 6 h)+L-Cys (24 μg/mL, 1 h) upon 660 nm laser irradiation (50 mW/cm², 3 min) using DCF-DA kit. **c** Mitochondrial membrane potential of cells treated with PBS buffer, ZNNPs@FA (12 h, 50 μg/mL), ZNNPs@FA (12 h, 50 μg/mL)+660 nm (50 mW/cm², 3 min) using JC-1 dye. **d** Viability of HCT116 cells after 12 h of treatment with ZNNPs@FA (0, 0.1, 1, 10, 25, 50, and 100 μg/mL), or pretreated with ZnCl2 (300 μM) for 10 min measured by MTT assay. Data are presented as mean ± s.d. (n = 3). Statistical differences were analyzed by Student's two-sided t-test. Source data are provided as a Source Data file.

among all these groups was determined during the treatment (Fig. 8e). These results strongly prove that this H2S-activatable probe ZNNPs@FA can not only suppress the tumor growth by scavenging the endogenous H2S, but also exhibits high antitumor activity under 660 nm laser irradiation, which is further validated by the hematoxylin and eosin (H&E) staining of tissue slices extracted from main organs (e. g. heart, liver, spleen, lung, and lung) and tumors in mice with various treatments. Likewise, both TUNEL and Ki67 immunohistochemical staining revealed that significant DNA damage and low malignancy presented in ZNNPs@FA and ZNNPs@FA + 660 nm-treated tumor tissues (Fig. 8f and Supplementary Fig. 33), indicative of effective tumor suppression. Moreover, we investigated the blood circulation, blood biochemistry, biodistribution, and pharmacokinetic behavior of our probes in healthy mice by real-time measuring the probe contents in blood. After calculation, ZNNPs and ZNNPs@FA have blood circulation time with a $T_{1/2}$ of 0.43 h and 0.65 h, respectively (Supplementary Fig. 34). Their biodistribution and excretion results indicate that both probes mainly accumulated in the liver and lung at 3 h post-injection. With the time elapses, the probes were slowly excreted from the body through the organs of the liver, lung, and kidney. Less than 5% ID$^{-1}$ of the initial probes in the major organs was retained at 48 h post-injection (Supplementary Figs. 35–38), further demonstrating our probes possess excellent biological safety. Collectively, all of the above evidences firmly demonstrate that the unexpected therapeutic effect endowed by endogenous H2S depletion combined with activated PDT makes this probe very promising for efficient cancer therapy.

## Discussion

In summary, we developed a smart, H2S-responsive and depleting nanoplatform ZNNPs and demonstrated their utility for quantitative visualization and depletion of endogenous H2S combined with activated PDT effect for colorectal tumor treatment. Our studies show that nanoprobe ZNNP with unexpected ratiometric PA responsiveness towards H2S at 680 nm and 900 nm offers deeper imaging penetration and improved sensitivity for non-invasive and quantitative detection of H2S level in mice, which will facilitate the study of H2S-associated liver dysfunctions, brain injury, and tumors in vivo. More notably, the depletion of H2S combined with activatable PDT of ZNNP@FA under NIR irradiation significantly improved the therapeutic efficacy of colorectal tumors in living mice. We thus believe that the design and application of this smart H2S-responsive nanoplatform may open an avenue for precise diagnosis and efficient intervention of H2S-related diseases.

## Methods

**Materials**. All chemicals were purchased from Saan chemical technology (Shanghai) unless otherwise stated and used without further purification. mPEG5000-PCL3000 and mPEG5000-PCL3000-FA were bought from ShangHai ToYongBio Tech. Inc. N-Hydroxy succinimide, DL-PROPARGYLGLYCINE (PAG), L-Cys, Metformin, Type IV collagenase, GAPDH Activity Assay Kit and Triethylamine were purchased from Sigma-Aldrich. 1,3-Diphenylisobenzofuran and p-Nitrobenzoyl chloride were purchased from TCI (Shanghai). Hydrogen Sulfide (H2S) Colorimetric Assay Kit was purchased from Elabscience Biotechnology. Mito-Tracker Green and MTT were purchased from Beyotime Biotechnology. Agilent Seahorse XF Cell Mito Stress Test Kits and Agilent Seahorse XF Glycolysis Stress Test Kits were purchased from Agilent Technologies Inc. DCFH-DA was purchased from Solarbio (Beijing). HCT116 cells and human emborynic

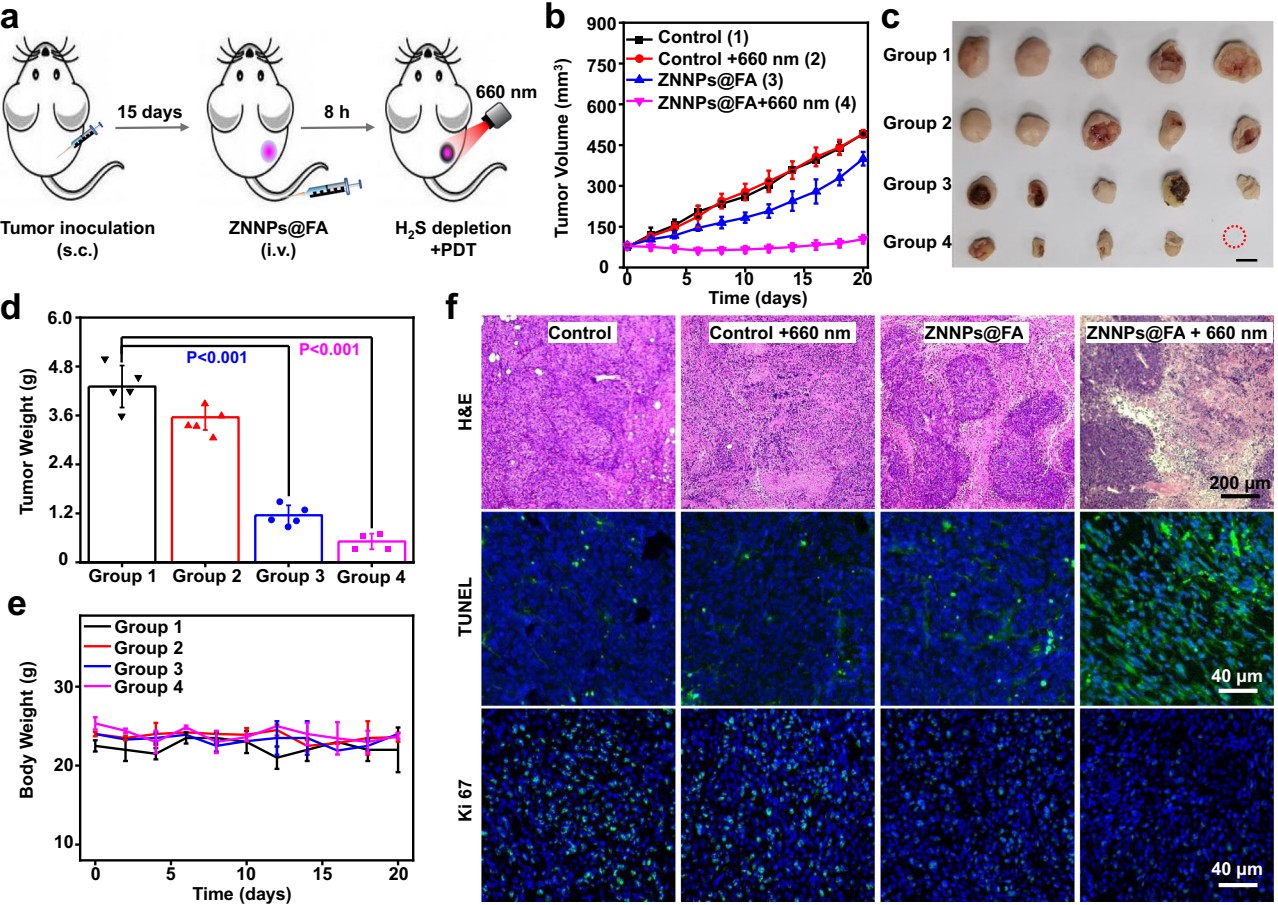

**Fig. 8 In vivo PDT of HCT116 subcutaneous tumors with ZNNPs@FA. a** Schematic for tumor construction and PDT treatment. **b** Tumor growth profiles of the mice treated by PBS buffer (200 μL) or ZNNPs@FA (10 mg/kg), and mice in the second and fourth groups were treated with 660 nm laser irradiation (50 mW/cm$^2$, 3 min) at 16 h. **c** Tumor images and **d** tumor weight of each group at the end of treatment. **e** Body weight following above treatments. **f** H&E, Tunnel, and ki67 staining images of tumor slices 2 days after indicated treatments. Data are presented as mean ± s.d. ($n = 5$). Statistical differences were analyzed by Student's two-sided t-test. Source data are provided as a Source Data file.

kidney 293 (HEK293) cells were purchased from Stem Cell Bank, Chinese Academy of Sciences (Shanghai, China). All cell lines were routinely tested for mycoplasma contamination, and cells were authenticated by Short Tandem Repeat test.

**Instrumentation**. HPLC profiles were acquired by using 1260 high-performance liquid chromatography (Agilent). Mass spectra were recorded on a Q Exactive™. NMR spectra were recorded on a Bruker Advance III 400 MHz spectrometer at 25 °C. UV absorption spectra were taken on a UV spectrometer (UV-3600, Shimadzu). The fluorescence spectra were measured on an Edinburgh FLS980 spectrophotometer. DLS were performed on a particle size analyzer (Nano ZS90, Malvern). PA imaging was performed with a Multispectral Optoacoustic Tomography scanner (MSOT, iThera medical, Germany). The fluorescence images in vivo were acquired using an IVIS spectrum imaging system (PerkinElmer) and NIR-II imaging system (Suzhou Yingrui Optical Technology Co., LTD). The confocal images were captured on a fluorescence microscope (FV1200, Olympus). OCR and ECAR were measured by a Seahorse Bioscience instrument (XF24, Agilent).

**Synthesis of ZM1068-ketone**. ZM1068-ketone was synthesized according to the previously reported method[42–45]. Briefly, N-hydroxysuccinimide (334 mg, 2.9 mmol) and triethylamine (0.5 mL) were mixed in 10 mL anhydrous DMF. ZM1068 (644 mg, 1 mmol), dissolved in 6 mL of anhydrous DMF, was then added drop by drop under agitation. The mixture was subsequently stirred at 25 °C under a nitrogen atmosphere for 3 h. Finally, ZM1068-ketone (488 mg, 78%) was obtained after HPLC purification. $^1$H NMR (600 MHz, DMSO-d6): δ 8.41 (d, $J = 13.2$ Hz, 2H), 8.03 (d, $J = 7.5$ Hz, 2H), 7.86 (d, $J = 8.1$ Hz, 2H), 7.79-7.71 (m, 2H), 7.43 (dd, $J = 8.1$, 7.4 Hz, 2H), 7.30 (d, $J = 8.3$ Hz, 2H), 6.82 (d, $J = 7.3$ Hz, 2H), 6.15 (d, $J = 13.3$ Hz, 2H), 4.60 (s, 2H), 4.08–3.92 (m, 4H), 2.84–2.70 (m, 4H), 2.03–1.92 (m, 3H), 1.78-1.70 (m, 4H). $^{13}$C NMR (151 MHz, DMSO-d6): δ 174.90, 167.37, 147.53, 146.49, 143.05, 139.21, 131.93, 130.83, 129.75, 129.37, 126.87, 123.22, 120.42, 116.94, 106.01, 100.82, 54.44, 40.74, 31.58, 24.92, 22.52. MS (ES + ): $m/z$ calcd for $C_{40}H_{36}N_2O_5$ 624.2624. Found 623.3261.

**Synthesis of ZM1068-NB**. ZM1068-ketone (100 mg, 0.16 mmol) was dissolved into 10 mL anhydrous dichloromethane. p-nitrobenzoyl chloride (89 mg, 0.48 mmol) dissolved in 5 mL of anhydrous dichloromethane was then added drop by drop under agitation. After 12 h stirring, ZM1068-NB (68 mg) was obtained after HPLC purification with a yield of 54.9%. $^1$H NMR (600 MHz, CD$_3$OD) δ 8.59, 8.02, 8.00, 7.81, 7.80, 7.48, 7.46, 7.26, 7.09, 7.08, 6.62, 6.59, 6.17, 4.87, 4.18, 3.67, 3.31, 2.85, 2.76, 2.09, 1.89, 1.88, 1.86, 1.73, 1.29, 1.26, 0.01. $^{13}$C NMR (151 MHz, CD$_3$OD): δ 164.12, 159.02, 154.07, 153.07, 141.64, 138.97, 134.42, 132.76, 132.32, 131.14, 130.48, 130.34, 129.80, 127.84, 127.63, 126.02, 125.55, 124.29, 111.07, 108.31, 70.04, 49.23, 49.08, 48.94, 48.80, 48.66, 48.52, 48.37, 42.33, 41.14, 29.55, 29.31, 25.54, 21.94. MS (ES +): $m/z$ calcd for Chemical Formula: $C_{47}H_{40}N_3O_8^+$ 774.2810. Found 623.3261.

**Fabrication of ZNNPs Nanoparticles**. ZM1068-NB (1 mg) and mPEG$_{5000}$-PCL$_{3000}$ (4 mg) were separately dissolved in 100 μL of dimethyl sulfoxide to give stock solutions. The solution of ZM1068-NB was added dropwise into the mPEG$_{5000}$-PCL$_{3000}$ solution during 10 min under ultrasonication. The mixture was then added into 4 mL of distilled water dropwise under ultrasonication during 10 min. The excessive reagents were purified by ultrafiltration (MWCO = 50 kDa) to afford the desired ZNNPs nanoparticles. The hydrodynamic diameter was measured on Zetasizer ZS90 (Malvern, UK). The morphology was characterized by TEM (Tecnai G20).

**Fabrication of ZNNPs@FA Nanoparticles**. ZNNPs@FA nanoparticles were prepared following above the same method, except that 0.2 mg mPEG$_{5000}$-PCL$_{3000}$-FA was added into the solution containing 4 mg of mPEG$_{5000}$-PCL$_{3000}$. The hydrodynamic diameter was measured on Zetasizer ZS90 (Malvern, UK). The morphology was characterized by TEM (Tecnai G20).

**Reactivity of ZM1068-NB towards H$_2$S**. To test the reactivity of ZM1068-NB toward H$_2$S, 1 mg of ZM1068-NB was dissolved in 1 mL DMSO to obtain a stock

solution of 1 mg/mL. Different concentrations of NaHS aqueous solution (0, 15, 30, 60, 90, 120, 250, and 500 μM) were prepared, 20 μL ZM1068-NB stock solution was then added into 1.98 mL hydrogen sulfide solutions with different concentrations of NaHS and incubated at 37 °C for 10 min. The reaction was subsequently characterized by UV-vis absorption spectra before and after addition of ZM1068-NB. The photoacoustic signals of the assays ranging from 0 and 60 μM were also measured. In order to characterize the products of the reaction between ZM1068-NB and $H_2S$, 20 μL of ZM1068-NB stock solution (1 mg/mL) was added into 1.98 mL of NaHS solution (60 μM). The mixture was incubated at 37 °C for 10 min and analyzed by HPLC. The products at 7.2 min and 13.2 min were purified by HPLC and analyzed by Q Exactive™.

**Reactivity of ZNNPs towards $H_2S$.** To investigate the reactivity of ZNNPs toward $H_2S$, 1.4 mg ZNNPs was dissolved in 1 mL of PBS buffer to obtain a stock solution of 1.4 mg/mL. Different concentrations of NaHS solutions ranging from 0 to 500 μM were prepared, 20 μL of ZNNPs stock solution was then added into 1.98 mL of NaHS solution with different concentrations and incubated at 37 °C for 10 min. The assays were analyzed by UV-vis spectroscopy, fluorescence spectra, IVIS, NIR-II imaging system, and MSOT immediately. Besides, the absorption variation of ZNNPs stock solution (20 μL, 1.4 mg/mL) with 1.98 mL $H_2S$ solution (100 μM) was monitored by UV-vis spectroscopy for 10 min.

**Selectivity of ZNNPs towards $H_2S$.** To evaluate the specificity and selectivity of ZNNPs toward $H_2S$, 30 μL of ZNNPs (1.4 mg/mL) in PBS buffer (pH 7.4) were added into 1.97 mL of different biologically relevant species (100 μM NaHS, 1 mM $SCN^-$, 1 mM $NO_3^-$, 1 mM $SO_4^{2-}$, 1 mM $K^+$, 1 mM $Ca^{2+}$, 1 mM $Na^+$, 1 mM $Mg^{2+}$, 1 mM $NO_2^-$, 1.25 mM VC, 10 mM GSH, 1 mM $H_2O_2$, 1 mM $CO_3^{2-}$, 1 mM $S_2O_3^{2-}$, 1 mM $SO_3^{2-}$, 1 mM $AC^-$) and incubated at 37 °C for 10 min. The fluorescence spectra from 640 to 850 nm and UV-vis absorption spectra from 550 to 1100 nm were recorded, respectively. The fluorescence images were acquired on an IVIS Spectrum imaging system with $\lambda_{ex}/\lambda_{em} = 640/720$ nm. The PA images of 680 nm and 900 nm were acquired on Multispectral Optoacoustic Tomography scanner.

**Cell culture.** HCT116 cells were cultured in DMEM medium (10% FBS, 100 units/mL penicillin and 100 units/mL streptomycin) and cultured in 5% $CO_2$ incubator at 37 °C. HEK293 cells were cultured in MEM medium (10% FBS, 100 units/mL penicillin and 100 units/mL streptomycin) and cultured in 5% $CO_2$ incubator at 37 °C.

**Cytotoxicity test of ZNNPs or ZNNPs@FA.** HCT116 cells or HEK293 cells was seeded into 96 well plates at density of $4 \times 10^3$ cells per well and grown for 24 h. And then cells were incubated with different concentrations (0, 0.1, 1, 10, 25, 50, 100 μg/mL) of ZNNPs or ZNNPs@FA for 24 h. The relative cell viabilities were finally determined by the standard MTT assay.

**Detection of endogenous $H_2S$ level in HCT116 cells.** HCT116 cells (~$5 \times 10^4$) were seeded in confocal dishes, and the cells adhered to the bottom after 12 h. In order to evaluate the responsiveness of ZNNPs toward $H_2S$ in HCT116 cells, seven groups of cells were parallelly set up. The second group was treated with $ZnCl_2$ (300 μM, in DMEM medium) for 10 min; the fourth group was treated with L-Cys (200 μM, in DMEM medium) for 1 h; the fifth group was treated with PAG (50 mg/mL, in DMEM medium) for 30 min to inhibit the CSE activity in HCT116 cells and treated with L-Cys (200 μM, in DMEM medium) for 1 h; the sixth group was treated with LPS (1 μg/mL, in DMEM medium) for 6 h to increase the expression of CSE in HCT116 cells and treated with L-Cys (200 μM, in DMEM medium) for 1 h; the seventh group was treated with LPS (1 μg/mL, in DMEM medium) for 6 h to increase the expression of CSE in HCT116 cells, and treated with PAG (50 mg/mL, in DMEM medium) for 30 min, then treated with L-Cys (200 μM, in DMEM medium) for 1 h. All cells in the seven groups were treated with ZNNPs (20 μg/mL) and incubated for 4 h. The third group needs to be treated with NaHS for another 1 h. After each step of the above treatment, it is necessary to wash it with PBS buffer before the next operation. Finally, cells were placed in fresh DMEM medium for confocal imaging to collect fluorescence signals activated by $H_2S$ in HCT116 cells. The cell images were analyzed using the ImageJ software. Finally, each group of cells were collected and evenly dispersed into PBS for photoacoustic signals detection at 680 nm and 900 nm.

**Animals and tumor models.** All animal experiments were conducted according to the Guidelines for the Care and Use of Laboratory Animals of Soochow University and were approved by the Animal Ethics Committee of the Soochow University Laboratory Animal Center (Suzhou, China). For subcutaneous tumor model, the tumor weight should not exceed 10% of normal body weight, or 13 mm in diameter (about 1098.5 mm$^3$ in volume). The six-week-old Balbc-Nu male mice with body weights of 14-16 g were purchased from Chang Zhou Cavensla Experimental Animal Technology Co. Ltd. The mice were housed under standard conditions ($25 \pm 3$ °C, $60\% \pm 10\%$ relative humidity) with 12 h light/dark cycle. The tumors were grafted by injection of $3 \times 10^6$ HCT116 cells in 50 μL of PBS into the back of

each mouse. The tumor length and width were measured with caliper and tumor volumes were calculated by (length×width$^2$)/2. In all experiments, the tumor volumes did not exceed the maximal tumor size/burden permitted by the Animal Ethics Committee of the Soochow University.

**NIR/PA imaging of endogenous $H_2S$ in mouse liver.** To evaluate the response of ZNNPs to $H_2S$ in liver, Balbc-Nu mice were intraperitoneally injected with saline (100 μL), L-Cys (6 mM, 100 μL) or PAG (2 mg/mL, 100 μL) to develop the mice with different levels of $H_2S$ expression in liver. After 30 min, ZNNPs (1 mg/mL, 200 μL) was i.v. injected into mice. Images at indicated time point were acquired on IVIS (*Ex*: 640 nm, *Em*: 720 nm) and NIR-II imaging system (*Ex*: 808 nm, *Em*: 1070 nm). PA images (680 nm & 900 nm) at indicated time point were acquired on Multispectral Optoacoustic Tomography scanner.

**Establishment of in vivo quantitative detection method for $H_2S$.** To develop a quantitative method for endogenous $H_2S$ in vivo, Balbc-Nu mice ($18 \pm 0.5$ g) were intraperitoneally injected with 100 μL of L-Cys (0, 1, 2, 3, 4, and 5 mM) to develop the mice with different levels of $H_2S$ expression in liver. After 30 min, ZNNPs (1 mg/mL, 200 μL) was i.v. injected into mice through tail vein. Images at indicated time point were acquired on IVIS (*Ex*: 640 nm, *Em*: 720 nm) and NIR-II imaging system (*Ex*: 808 nm, *Em*: 1070 nm). PA images (680 nm & 900 nm) were acquired on Multispectral Optoacoustic Tomography scanner, and the averaged PA signal intensity in ROI region of the liver at 90 min was measured by iThera MSOT imaging software. Finally, the livers of each group ($n = 3$) were dissected and homogenized, and the actual concentration of $H_2S$ in the livers was measured using the $H_2S$ Colorimetric Assay Kit. Then $PA_{680}/PA_{900}$ was fitted with the actual concentrations of $H_2S$ in the liver to give a quantitative equation for endogenous $H_2S$ in living mouse.

**Quantification of $H_2S$ level in mouse liver.** Balbc-Nu mice ($18$ g $\pm 0.5$ g) were intraperitoneally injected with metformin (0, 1, 3, and 5 mg in 100 μL PBS buffer) for 7 consecutive days. ZNNPs (1 mg/mL, 200 μL) was i.v. injected into mice through tail vein. PA images of mouse livers were acquired at 680 nm and 900 nm on Multispectral Optoacoustic Tomography scanner, and the averaged PA signal intensity in ROI region of the liver at 90 min was measured by the iThera MSOT imaging software. Then $PA_{680}/PA_{900}$ was substituted into the quantitative equation

$$y = -0.90573x^2 + 5.30757x - 2.23108$$

to obtain the concentration of $H_2S$ in mice liver. Finally, the livers of each group ($n = 2$) were dissected and homogenized, and the actual concentration of $H_2S$ in the livers was measured using the $H_2S$ Colorimetric Assay Kit.

**ICH model and quantification of $H_2S$ in injured mouse brain.** ICH model was developed according to previously reported method[58]. The ICR mice were anesthetized with 4% chloral hydrate solution, then type IV collagenase 0.5 μL (2.5 mg/mL) unilaterally into the left striatum (1 mm anterior and 2.0 mm lateral of the bregma, 3.5 mm in depth) by craniotomy. The collagenase was delivered over 5 min and the needle stayed in place for an additional 5 min. After 24 h, ZNNPs (1 mg/mL, 200 μL) was i.v. injected into mice through tail vein. Images at indicated time point were acquired on IVIS (*Ex*: 640 nm, *Em*: 720 nm) and NIR-II imaging system (*Ex*: 808 nm, *Em*: 1070 nm). PA images of mice liver were acquired at 680 nm and 900 nm on Multispectral Optoacoustic Tomography scanner, and the averaged PA signal intensity in ROI region of a cerebral hemorrhage at 90 min was measured by iThera MSOT imaging software. Then $PA_{680}/PA_{900}$ was substituted into the quantitative equation

$$y = -0.90573x^2 + 5.30757x - 2.23108$$

to give the concentration of $H_2S$ at the cerebral hemorrhage site. Finally, fluorescence images of mice brain sections were obtained on IVIS (*Ex*: 640 nm, *Em*: 720 nm) and NIR-II imaging system (*Ex*: 808 nm, *Em*: 1070 nm).

**In Vitro HCT116 cell proliferation assays.** HCT116 cells were seeded at the density of 3000 cells per well in xCELLigence plates treated with ZNNPs@FA (20 μg/mL) for 4 h or $ZnCl_2$ (300 μM) for 10 min when the cells reached a confluent state. The proliferation was monitored by xCELLigence system (ESSEN) for 48 h.

**Cell scratch test.** HCT116 cells in 6-well plate were treated with ZNNPs@FA (20 μg/mL) for 4 h or $ZnCl_2$ (300 μM) for 10 min when the cells reached a confluent state. A single scratch was made by 200 μL pipette tip. The cells were then incubated with FBS - free culture medium. Images of the scratches were captured at 0 and 24 h with Olympus IX73 inverted microscope (10×). The width of the scratch was analyzed by image J software.

**Determination of GAPDH activity in HCT116 cells.** HCT116 cells were incubated with different concentrations of ZNNPs@FA (0, 25, 50, 75, 100 μg/mL) for 24 h. The cells in each group were then homogenized for GAPDH activity detection using GAPDH Activity Assay Kit.

**Determination of H$_2$S level in HCT116 cells**. HCT116 cells were incubated with different concentrations (0, 25, 50, 100 μg/mL) of ZNNPs@FA for 24 h. The cells in each group were homogenized for H$_2$S relative concentration determination using Hydrogen Sulfide (H$_2$S) Colorimetric Assay Kit.

**Mitochondrial membrane potential assay**. HCT116 cells ($\sim5 \times 10^4$) were seeded in confocal dishes, and the cells adhered to the bottom after 24 h. All cells in the five groups were treated with ZNNPs@FA (0, 0.1, 1, 10, and 50 μg/mL) and incubated for 24 h. The cells were then washed by PBS buffer. The mitochondrial membrane potential of HCT116 cells was eventually measured by JC-1 dye (1 μM).

**Glycolysis and mito stress tests**. HCT116 cells (50,000 cells per well, 100 μL medium) were cultured in SeahorseXFe24 cell culture plate. The plate was placed in an ultra-clean table for 1 h, the cells were then cultured in an incubator containing 5% CO$_2$ at 37 °C. After 6 h, 150 μL medium was added into each well and cultured overnight. HCT116 cells were subsequently incubated with ZNNPs@FA (50 μg/mL or 100 μg/mL) or ZnCl$_2$ (20 μg/mL) for 6 h, the mitochondrial stress of HCT116 cells were measured in special medium (2 mM glutamine) with Glycolysis Stress Test Kits on Agilent Seahorse XFe24 (oligomycin: 30 μM, 56 μL, FCCP: 2.5 μM, 62 μL, Rot/AA: 5 μM, 69 μL). In parallel, the mitochondrial stress of the HCT116 cells were also measured in a special medium (1 mM pyruvate, 2 mM glutamine, 10 mM glucose) with Seahorse XF Cell Mito Stress Test Kits on Agilent Seahorse XFe24 (glucose: 100 mM, 56 μL, oligomycin: 10 μM, 62 μL, 2-DG: 500 mM, 69 μL).

**H$_2$S depletion-based suppression of subcutaneous HCT116 tumor**. 5-6 weeks old male Balbc-Nu mice were inoculated with HCT116 cells ($8 \times 10^6$ cells) via a subcutaneous injection into their right and left thigh. Two weeks later, ZNNPs@FA (20 μL, 1.4 mg/mL) (left) and ZNNPs@FA (20 μL, 1.4 mg/mL)/NaHS (20 μL, 5 μM) were intratumorally injected into left and right side of tumors, respectively, every two days, and the tumor volume was recorded every two days. After 10 days, the tumors were removed and sliced for immunofluorescence staining using antibodies against CD-31 and VEGF.

**Photodynamic effect of ZNNPs@FA in solution**. Different concentrations (0, 37, 75, 150, and 300 μM, 2 mL) of NaHS solutions were mixed with ZNNPs@FA (60 μg) and DPBF (43 μg) for singlet O$_2$ detection. These samples were irradiated by 660 nm laser (50 mW/cm$^2$), and the UV absorbance at 415 nm was measured at 0, 30, 60, 120, 180, 240, 300, and 360 s.

**Photodynamic effect of ZNNPs@FA in HCT116 cells**. HCT116 cells ($\sim5 \times 10^4$) were seeded in confocal dishes, and then adhered to the bottom after 12 h. Five groups of cells including four experimental and one control groups were set up. The second group of cells was treated with NaHS for another 1 h. The third group was treated with ZnCl$_2$ (300 μM, in DMEM medium) for 10 min; the fourth group was treated with L-Cys (200 μM, in DMEM medium) for 1 h; the fifth group was treated with PAG (50 mg/mL, in DMEM medium) for 30 min and treated with L-Cys (200 μM, in DMEM medium) for 1 h. All cells in five groups were washed and incubated with ZNNPs@FA (20 μg/mL) for 4 h. The cells were then washed by PBS buffer for three times, and incubated with DCFH-DA (10 μM, 1 mL) for 20 min. Finally, the cells were washed using serum-free medium for three times, it was further exposed to 660 nm (50 mW/cm$^2$) for 3 min followed by assessing the production of singlet O$_2$. under the fluorescence microscope.

**H$_2$S-activated PDT of ZNNPs@FA in HCT116 cells**. HCT116 or HEK293 cells were seeded into 96 well plates at density of $8 \times 10^3$ cells per well and grown for 24 h. The third group HCT116 cells were incubated with ZnCl$_2$ (300 μM) for 10 min, the cells were incubated with ZNNPs@FA (10 μg/mL) for 24 h. Finally, the cells were washed by PBS buffer for three times and subsequently irradiated by 660 nm (50 mW/cm$^2$) laser for 3 min. The cell viabilities were finally determined by the standard MTT assay.

**Co-localization of ZNNPs@FA with mitochondria**. HCT116 cells ($\sim5 \times 10^4$) were seeded in confocal dishes, and adhered to the bottom after 24 h. All cells were washed and incubated with ZNNPs@FA (20 μg/mL) for various time points, then washed by PBS buffer for three times. Finally, cells were incubated with Mito-Tracker Green (100 nM, 1 mL) for 30 min and washed by PBS buffer for fluorescence microscope imaging.

**Mitochondrial membrane potential measurement after PDT**. HCT116 cells ($\sim5 \times 10^4$) were seeded in confocal dishes, and adhered to the bottom after 12 h. The cells were then treated with PBS buffer (12 h), or ZNNPs@FA (12 h, 50 μg/mL), or ZNNPs@FA (12 h, 50 μg/mL)+660 nm (50 mW/cm$^2$, 3 min). After 24 h, the cells were washed by PBS buffer. The mitochondrial membrane potential were determined by using JC-1 (1 μM). Furthermore, the morphology of cellular mitochondria was also observed by Cryo-TEM.

**In vivo NIR imaging of HCT116 tumors**. HCT116 xenograft tumor-bearing mice were intravenously injected with ZNNPs@FA (1.4 mg/mL, 200 μL) through tail vail. The mice were then anesthetized with 3% isoflurane mixed with air and imaged by IVIS Spectrum system, NIR-II fluorescent imaging system and MSOT at different time points.

**In Vivo PDT of HCT116 tumors**. HCT116 xenograft tumor-bearing mice were randomly grouped (five mice per group). Each mouse was intravenously injected with PBS buffer (200 μL) or ZNNPs@FA (1.4 mg/mL, 200 μL) through tail vein. After 16 h, the tumors were irradiated by 660 nm laser (50 mW/cm$^2$) for 3 min. The tumor size was measured every two days, and the mice were sacrificed on the 20th day. The tissue slices of heart, lung, spleen, liver, kidney, and tumor were analyzed by H&E staining. The slices of the tumor were further analyzed by TUNEL and Ki67 immunofluorescence staining.

**Data analysis**. Average area intensities in cells were given by a fluorescence microscope (FV1200, Olympus). Average area intensities in mice were given by the Living Image software of PerkinElmer IVIS. All data were analyzed and calculated with Microsoft Excel 2019 software (Microsoft), and the statistical differences were analyzed by a two-tailed student's test. All statistical data were presented as means ± SD. All statistical graphs and fluorescent spectra were performed using Origin 2020.

**Reporting summary**. Further information on research design is available in the Nature Research Reporting Summary linked to this article.

## Data availability
The main data supporting the results in this study are available within the Article, its Supplementary Information and Source Data File. Source data are provided with this paper.

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

## Acknowledgements

This work was supported by grants from the Training Program of the Major Research Plan of the National Natural Science Foundation of China (91959123), the National Natural Science Foundation of China (22077092), the Key Research and Development Program of Social Development of Jiangsu Province (BE2018655), the Open Project Program of the State Key Laboratory of Radiation Medicine and Protection (GZK1202132 and GZK1202017) and a project funded by the Priority Academic Program Development of Jiangsu Higher Education Institutions. H. Shi would like to thank prof Wang Qiangbin in Suzhou Institute of Nano-Bionics (SINANO) for his help and support in NIR-II fluorescence cell imaging.

## Author contributions

H.S. conceived and designed the experiments. Y.Z., J.F., S.Y., Q.M., Y.Z., C.C., A.W., Y.F., and J.L. performed the experiments. Y.Z., M.Z. and H.S. analyzed the data and supervised the project. Y.Z., and H.S. wrote the manuscript and all authors discussed the results and commented on the manuscript.

## Competing interests

The authors declare no competing interests.
