## [Peer Review File · Nature Communications]

Reviewers' Comments:

Reviewer #1:

Remarks to the Author:

In this manuscript, the authors developed an intelligent, H₂S-responsive and depleting nanoplatform ZNNPs by encapsulating NIR-II fluorophore ZM1068-NB into amphiphilic polymer mPEG5000-PCL3000 for in vivo quantitatively visualization of endogenous H₂S through the ratiometric photoacoustic (PA680/PA900) signal variation. The feasibility and usefulness of this nanoplatform for quantitative imaging and depletion of H₂S inducing cell proliferation suppression were strongly demonstrated in acute hepatotoxicity, cerebral hemorrhage and colorectal tumor models in living mice. Hence, this study would provide a useful and universal means for quantitatively studying the malignant impacts of tumor-associated biomolecules in living system. The topic is interesting, and the data are solid to support the conclusions. This manuscript can be recommended for publication in Nature Communication after addressing the following minor issues:

1. We are wondering whether the ratiometric photoacoustic signal (PA680/PA900) of ZNNPs towards H₂S could be interfered by some external factors, e. g. pH, viscosity, etc. Please perform experiments to validate it.
2. Although both ZNNPs and ZNNPs@FA showed good H₂S responsiveness in vivo, their stability in serum-containing solution is lacking, which should be studied.
3. The data show that ZNNPs have good responsiveness towards H₂S. How about the detection limit of ZNNPs toward H₂S. What is the expression level of H₂S in liver or colorectal tumor normally?
4. Toward in vivo applications, the blood circulation half-life of ZNNPs and ZNNPs@FA should be also studied, and the cytotoxicity of ZNNPs and ZNNPs@FA in mice should carried out, for example, blood hemolysis and hemanalysis.
5. According to the results in Figure 1d, the FL1070 is gradually decreased with the increasing concentration of H₂S used, why does the FL1070 almost kept constant for NIR imaging of H₂S in livers (Fig 3b)? Please make a clarification on it.
6. Some formatting and typo errors present in the manuscript. Please Go through the whole manuscript carefully.

Reviewer #2:

Remarks to the Author:

In the submitted manuscript the authors designed the H₂S-responsive and depleting nanoplatform ZNNPs to visualize and simultaneously scavenge the endogenous H₂S, together with the activated photodynamic effect, for synergistical enhanced colorectal cancer treatment. It is noted that numerous organic or inorganic nanoprobe have been explored for endogenous H₂S visualization and depletion including activated upconversion imaging, NIR-II imaging, NIR-II ratiometric imaging, and PA imaging (Nano Lett. 2018, 18, 6411; Biomaterials 2021, 275, 120918; Anal. Chem. 2021, 93, 9356-9363; Nano Lett. 2021, 21, 4606, Angew.Chem., Int. Ed. 2018, 57, 3626; Angew. Chem. Int. Ed. 2018, 57, 15782; Chem. Sci., 2019, 10, 1193; Theranostics2019, 9, 77 et al). And the H₂S activated NIR-II nanoprobe and therapy have also been extensively reported in the literature and thus the novelty of this work is lacking. Therefore, in my opinion, this manuscript is suitable for publication in more other specialized journal instead of Nature Communications.

Some detailed comments are listed as follows:

1. Subcutaneous tumor is not sufficient for present work, and the in situ tumor model should be used for the imaging and therapy.
2. Biological characterization is too rough. For example, statistical analysis, cell toxicity, pathological section, blood biochemical analysis, pharmacokinetics and survival curve are missed.
3. As the author mentioned, the ZNNPs can be used for brain tissue imaging. How did they broken the blood-brain barrier? The authors should give detailed evidence and explanation.
4. The NIR-II cellular imaging also should be performed for H₂S response.
5. The accumulation of nanoparticles in tumors was limited, which will affect the therapeutic effect.
6. What is the detection limitation of the explored nanoprobe for H₂S?
7. Authors also missed the QY of nanoprobe.

8. In Figure S20, there are obvious interference signal from NIR-II imaging and NIR imaging. It seems the specific diagnosis of HCT116 tumor is very poor. And the authors should give the detailed explanation on the interference signal in the activated nanoprobe.

Reviewer #3:

Remarks to the Author:

In the article "Quantitatively Visualizing and Depleting Endogenous H₂S Combined with Activated Photodynamic Effect for Synergistical Enhanced Colorectal Cancer Therapy", the authors report a smart, H₂S-responsive and depleting nanoplatform (ZNNPs) to quantitatively and real-time imaging endogenous H₂S for early diagnosis and treatment of H₂S-associated diseases (acute hepatotoxicity, cerebral hemorrhage, and colorectal tumor). This ZNNPs nanoplatform displayed unexpected NIR conversion (F1070→F720) and ratiometric photoacoustic (PA680/PA900) signal responsiveness towards H₂S and simultaneously depleted the mitochondrial H₂S level in cancer cells leading to remarkable reduction of glycolysis and severe mitochondrial damage, together with the activated photodynamic effect. Sufficient experiments and data were supplied to make this work nice. The nanoplatform is new and the application is important. Thus, I recommend its publication after a minor revision.

- 1) Reports with similar proposals and structures utilizing different chemistry and imaging mode have been reported in recent years (DOI: 10.1021/jacs.9b09181; 10.1021/jacs.5b04097 etc.). What is the big improvement of this work utilizing H₂S-responsive ZNNPs nanoplatform in comparison with those reports? Deeper discussion should be provided to make the manuscript persuasive.
- 2) Why author use two different cells (human embryonic kidney 293 (HEK293) cells and HCT116 cells) in this work?
- 3) In Fig. 3 and Fig. 7, please mark the scale bar in fluorescence and PA images.
- 4) The singlet oxygen (¹O₂) quantum yield of ZNNPs@FA should be provided.

Medical College of Soochow University

Reviewer #1 (Remarks to the Author):

Comments: In this manuscript, the authors developed an intelligent, H₂S-responsive and depleting nanoplatfrom ZNNPs by encapsulating NIR-II fluorophore ZM1068-NB into amphiphilic polymer mPEG₅₀₀₀-PCL₃₀₀₀ for in vivo quantitatively visualization of endogenous H₂S through the ratiometric photoacoustic (PA₆₈₀/PA₉₀₀) signal variation. The feasibility and usefulness of this nanoplatfrom for quantitative imaging and depletion of H₂S inducing cell proliferation suppression were strongly demonstrated in acute hepatotoxicity, cerebral hemorrhage and colorectal tumor models in living mice. Hence, this study would provide a useful and universal means for quantitatively studying the malignant impacts of tumor-associated biomolecules in living system. The topic is interesting, and the data are solid to support the conclusions. This manuscript can be recommended for publication in Nature Communication after addressing the following minor issues:

Response: Thanks for your affirmation and encouragement to us.

1. We are wondering whether the ratiometric photoacoustic signal (PA₆₈₀/PA₉₀₀) of ZNNPs towards H₂S could be interfered by some external factors, e. g. pH, viscosity, etc. Please perform experiments to validate it.

Reply: Thanks for the reviewer's great comments. In order to investigate the effect of pH on our sensing system, we mixed probe ZNNPs with different concentrations of NaHS ranging from 0 to 100 μM, respectively, in PBS buffer with various pH (5.0, 7.4, and 9.0) followed by PA signal measurement with MOST instrument. The results shown in Figure S13a indicate that the PAS₆₈₀/PAS₉₀₀ is gradually enhanced with the increasing concentrations of NaHS. However, under the same concentration of NaHS, the PAS₆₈₀/PAS₉₀₀ values of the assays in different pH buffer are quite similar, suggesting that the pH of the solution does not have apparent effect on the detection of H₂S using our probes. To further evaluate whether the buffer's viscosity affects the performance of our probe for H₂S detection, different proportions of glycerin were added into PBS buffer to prepare the working buffer in which probe ZNNPs and different concentrations of NaHS were mixed together. Similarly, no significant difference in PAS₆₈₀/PAS₉₀₀ was determined for each assay under the same condition (Figure S13b), revealing that the viscosity of the buffer has a negligible effect on the detection of H₂S for our sensing system.

Figure S13. The effect of pH (a) and viscosity (b) on the responsiveness of ZNNPs toward NaHS.

2. Although both ZNNPs and ZNNPs@FA showed good H₂S responsiveness in vivo, their stability in serum-containing solution is lacking, which should be studied.

Reply: Thanks for the reviewer's valuable comments and suggestions. To investigate the stability of ZNNPs and ZNNPs@FA in serum-containing solution, both probes were separately dispersed into 10% serum-containing PBS buffer and then stocked at 37°C for one week. The hydrodynamic sizes of ZNNPs and ZNNPs@FA were real-time monitored by DLS analysis. As shown in Figure S5c and S5d, no big change in terms of particle size for both probes was observed, suggesting that both ZNNPs and ZNNPs@FA have great stability in serum-containing aqueous solution.

Figure S5. The stability study of ZNNPs (c) and ZNNPs@FA (d) in PBS containing 10% serum at 37°C over one week.

3. The data show that ZNNPs have good responsiveness towards H₂S. How about the detection limit of ZNNPs toward H₂S. What is the expression level of H₂S in liver or colorectal tumor normally?

Reply: Thanks for your great comments. To determine the detection limit of ZNNPs toward H₂S, probe ZNNPs (14 μg/mL) was incubated with various concentrations of NaHS ranging from 0 to 500 μM for 10 min followed by detection of the fluorescence (FL₇₂₀) and ratiometric photoacoustic (PAS₆₈₀/PAS₉₀₀) signal. Figure S11c and S11d clearly shows that both FL₇₂₀ and PAS₆₈₀/PAS₉₀₀ were enhanced linearly with the increase of NaHS concentrations. According to the formula $LOD = 3\sigma/s$ (where LOD was the limit of detection, σ was the standard deviation of blank probe sample measurement, and s was the slope of the calibration pot, values are mean \pm SD, $n =$

3), the lowest detection limits of H₂S in fluorescence and ratiometric PAS₆₈₀/PAS₉₀₀ were eventually calculated to be ~4.7 μM and ~0.68 μM, respectively, which strongly demonstrates that the sensitivity of ratiometric PA is significantly superior to fluorescence in detection of H₂S.

Figure S11. Determination of the detection limit of ZNNPs toward H₂S in the fluorescence intensity at 720 nm and ratiometric PAS₆₈₀/PAS₉₀₀, respectively.

To know the expression level of H₂S in liver or colorectal tumor normally, we also searched and read a lot of literature, and found the concentration of H₂S in mouse liver is around 20 μM, and is ranging from 0.3 to 3.4 mmol L⁻¹ in mouse colorectal tumor. The related references are listed below:

1. Fiorucci, S.; Distrutti, E.; Cirino, G.; Wallace, J. L. The emerging roles of hydrogen sulfide in the gastrointestinal tract and liver. *Gastroenterology*. 131, 259-71 (2006).
2. Wang, R. Physiological implications of hydrogen sulfide: a whiff exploration that blossomed. *Physiol Rev*. 92, 791-896 (2012).
3. Feng, X, et al. Toxic effects of hydrogen sulfide donor NaHS induced liver apoptosis is regulated by complex IV subunits and reactive oxygen species generation in rats. *Environ Toxicol*. 35, 322-332 (2020).
4. Zhang, W, et al. Activatable nanoscale metal-organic framework for ratiometric photoacoustic imaging of hydrogen sulfide and orthotopic colorectal cancer in vivo. *Science China Chemistry*. 63, 1315-1322 (2020).
5. Wu, Y. C.; et al. Hydrogen sulfide lowers proliferation and induces protective autophagy in colon epithelial cells. *PLoS One*. 7, e37572 (2012).
6. An, L, et al. The In Situ Sulfidation of Cu₂O by Endogenous H₂S for Colon Cancer Theranostics. *Angew. Chem. Int. Ed*. 57, 15782-15786 (2018).
7. Szabo, C.; et al. Tumor-derived hydrogen sulfide, produced by cystathionine-beta-synthase, stimulates bioenergetics, cell proliferation, and angiogenesis in colon cancer. *Proc. Natl. Acad. Sci. USA*. 110, 12474-9 (2013).

4. Toward in vivo applications, the blood circulation half-life of ZNNPs and ZNNPs@FA should be also studied, and the cytotoxicity of ZNNPs and ZNNPs@FA in mice should be carried out, for example, blood hemolysis and hemanalysis.

Reply: Thanks for the reviewer's great comments. It is true that the blood circulation behavior of nanoprobe in body is one of important factors which should be investigated especially for in vivo biological applications. To this end, two groups of healthy mice were intravenously injected with ZNNPs and ZNNPs@FA, respectively. The blood samples were collected from the eyes at different time points for NIR fluorescence measurements. The results shown in Figure S32 indicate that both probes exhibited a short blood circulation time. After calculation, the main

pharmacokinetic parameters ($t_{1/2}$) are 0.43 h and 0.65 h for ZNNPs and ZNNPs@FA, respectively.

Figure S32. Blood clearance curve of ZNNPs ($t_{1/2} = 0.43$ h) (a) and ZNNPs@FA ($t_{1/2} = 0.65$ h) (b). Error bar indicates SD (n = 2). ZNNPs (10 mg/kg) and ZNNPs@FA (10 mg/kg) were i.v. injected into the mice. (100 μ L of blood were pretreated by adding 1 μ L of 50 mM NaHS solution and incubating at 37°C for 2 h).

To further evaluate the cytotoxicity of ZNNPs and ZNNPs@FA in mice, we performed the hemolysis analysis for both nanoprobes in mouse blood. The results shown in Figure S15 illustrated that no significant red cell damage was observed for both ZNNPs and ZNNPs@FA even at the concentration of 160 μ g/mL, implying that they don't have apparent hemolysis.

Figure S15. Hemolytic test of nanoprobes ZNNPs (a) and ZNNPs@FA (b) at different concentrations.

Meanwhile, we also carried out the blood routine analysis for both ZNNPs and ZNNPs@FA. The blood routine biochemical results shown in Figure S35 and S36 clearly indicate that no significant difference for all parameters (ALP, AST, ALT, UREA, CREA, RBC, WBC, PLT, MCV, MCH, MCHC, HGB, RDW, HCT, MPV, LYM, PCT, and PDW) associated with acute toxicity compared to the control blood was observed, again suggesting our probes have excellent biocompatibility.

Figure S35. Variations of blood biochemical and blood routine indexes of mice with i.v. injection of ZNNPs (10 mg/kg) at different time points in comparison with non-treated ones (Control). a, Alkaline phosphatase, ALP; aspartate aminotransferase, AST and alanine aminotransferase, ALT. b, urea nitrogen, UREA. c, creatinine, CREA. d, Red blood cells, RBC. e, White blood cells, WBC. f, Platelets, PLT. g, Mean corpuscular volume, MCV. h, Mean corpuscular hemoglobin, MCH. i, Mean corpuscular hemoglobin concentration, MCHC. j, Hemoglobin, HGB. k, Red cell distribution width, RDW. l, Hematocrit, HCT. m, Mean platelet volume, MPV. n, Lymphocyte, LYM. o, Plateletcrit, PCT. p, Platelet distribution width, PDW.

Figure S36. Variations of blood biochemical and blood routine indexes of mice with i.v. injection of ZNNPs@FA (10 mg/kg) at different days in comparison with non-treated ones (Control). a, Alkaline phosphatase, ALP; aspartate aminotransferase, AST and alanine aminotransferase, ALT. b, urea nitrogen, UREA. c, creatinine, CREA. d, Red blood cells, RBC. e, White blood cells, WBC. f, Platelets, PLT. g, Mean corpuscular volume, MCV. h, Mean corpuscular hemoglobin, MCH. i, Mean corpuscular hemoglobin concentration, MCHC. j, Hemoglobin, HGB. k, Red cell distribution width, RDW. l, Hematocrit, HCT. m, Mean platelet volume, MPV. n, Lymphocyte, LYM. o, Plateletcrit, PCT. p, Platelet distribution width, PDW.

5. According to the results in Figure 1d, the FL1070 is gradually decreased with the increasing concentration of H₂S used, why does the FL1070 almost kept constant for NIR imaging of H₂S in livers (Fig 3b)? Please make a clarification on it.

Reply: Thanks for your great question! Our in vitro results in Figure 1d are indeed shown that the FL₁₀₇₀ intensity of the probe gradually decreased with the increasing concentrations of H₂S. However, when the probes were intravenously injected into the mice via the tail vein, the probes accumulated in the mice liver gradually with the blood flow over time. In fact, the amount of the probes enriched in liver is dynamically changing due to the simultaneous occurring of clearance by the body. Hence, it is difficult to see a clear gradual decrease of FL₁₀₇₀ in the liver of living mice overall.

6. Some formatting and typo errors present in the manuscript. Please Go through the whole

manuscript carefully.:

Reply: We highly appreciated you pointed out our errors! We have carefully gone through the whole manuscript and made some revision accordingly.

Reviewer #2 (Remarks to the Author):

Comments: In the submitted manuscript the authors designed the H₂S-responsive and depleting nanopatform ZNNPs to visualize and simultaneously scavenge the endogenous H₂S, together with the activated photodynamic effect, for synergistical enhanced colorectal cancer treatment. It is noted that numerous organic or inorganic nanoprobes have been explored for endogenous H₂S visualization and depletion including activated upconversion imaging, NIR-II imaging, NIR-II ratiometric imaging, and PA imaging (Nano Lett. 2018, 18, 6411; Biomaterials 2021, 275, 120918; Anal. Chem. 2021, 93, 9356-9363; Nano Lett. 2021, 21, 4606, Angew. Chem., Int. Ed. 2018, 57, 3626; Angew. Chem. Int. Ed. 2018, 57, 15782; Chem. Sci., 2019, 10, 1193; Theranostics 2019, 9, 77 et al). And the H₂S activated NIR-II nanoprobe and therapy have also been extensively reported in the literature and thus the novelty of this work is lacking. Therefore, in my opinion, this manuscript is suitable for publication in more other specialized.

Response: We appreciate the reviewer's comments! As the reviewer said, it is true that many nanoprobes have recently been developed to visualize and deplete the endogenous H₂S for H₂S-activated NIR imaging and photothermal/photodynamic therapy of tumors both in vitro and in vivo. However, these previously reported studies mainly focused on the imaging of H₂S or activated tumor therapy. Differing from all of the previous studies, our novelty and innovation in this work can be summarized in two aspects: First, the key novelty of our study is that we not only developed a new H₂S-responsive NIR/ratiometric photoacoustic imaging probe for visualization of endogenous H₂S in vivo, more importantly, taking advantage of the ratiometric PA feature, we for the first time established a quantitative equation between ratiometric PA signal and the level of endogenous H₂S in living system. In light of this relationship, we successfully realized the noninvasive and quantitative visualization of endogenous H₂S in mouse liver, brain as well as colorectal tumor. Furthermore, we could achieve the quantitative evaluation of the drug hepatotoxicity in real-time manner by using our probes. Therefore, our approach may offer a powerful tool for studying the vital impacts of H₂S in related diseases; Second, we for the first time found our probe could also effectively deplete the endogenous H₂S in cellular mitochondria, which can cause significant ATP reduction and severe mitochondrial damage, combined with the H₂S-activated photodynamic effect, eventually leading to efficient suppression of colorectal tumor in living mice. Overall, our study would provide a new insight for achieving precise diagnosis and treatment of colorectal cancer.

Some detailed comments are listed as follows:

1. Subcutaneous tumor is not sufficient for present work, and the in situ tumor model should be used for the imaging and therapy.

Reply: Thanks for your suggestion! To be honest, our current work is mainly focusing on the proof of concept case study for developing a smart sensing platform to quantitatively assess the pathological progression of H₂S-associated diseases. I agree with the reviewer's opinion, the

experiments just based on subcutaneous tumors are not sufficient to fully validate the effectiveness and universality of our approach. Due to the complexity and time consumption of in situ tumor model, in the early study we mainly focus on validating the feasibility of our probes for quantitative detection of endogenous H₂S in subcutaneous tumors. To further explore the utility of our probes in vivo, we are currently building in situ mouse tumor models to deeply evaluate their potential for quantitative detection of endogenous H₂S in living mice. The results of the extensive studies will be reported later.

2. Biological characterization is too rough. For example, statistical analysis, cell toxicity, pathological section, blood biochemical analysis, pharmacokinetics and survival curve are missed.

Reply: Thanks for the reviewer's valuable comments! To make some biological characterization fully comprehensive, we performed further statistical analysis for some data in the revised manuscript. Additionally, to deeply evaluate the biocompatibility of probe ZNNPs and ZNNPs@FA, we also carried out their blood routine analysis. The blood routine biochemical results shown in Figure S35 and S36 indicate that no significant difference for all parameters (ALP, AST, ALT, UREA, CREA, RBC, WBC, PLT, MCV, MCH, MCHC, HGB, RDW, HCT, MPV, LYM, PCT, and PDW) associated with acute toxicity compared to the control blood was observed, suggesting our probes have excellent biocompatibility.

We next conducted a preliminary study of the biodistribution and pharmacokinetic behavior of both ZNNPs and ZNNPs@FA. The probes were i.v. injected into the healthy mice via the tail vein, and then the major organs including heart, liver, spleen, lung, and kidney were dissected and taken out for fluorescence imaging at 720 and 1070 nm. As shown in Figure S33, fluorescence signals of various organs indicated that the nanoprobe mainly accumulated in the liver and lung at 3 h post-injection. With the time elapses, both probes were slowly excreted from liver and lung into kidney, indicating the probes are gradually excluded from the body through the renal metabolism. The quantitative analysis of these fluorescence images showed almost all probes had been excreted from the body at 48 h post-injection. Furthermore, the organs at various time points were digested and lysed, and the supernatant were subjected for fluorescence measurement at 720 nm. The quantitative results in Figure S34 clearly indicate that approximately 16.5% of nanoprobe presented in every gram of liver of mice at 3 h post-injection, whereas less than 5.1% of them was still retained at 48 h post-injection, strongly implying that our probes can be rapidly metabolized by the body, and they have excellent biological safety. The survival rates of mice receiving different treatments were also analyzed. The results shown in Figure S30 indicate that ZNNPs@FA+660 nm group exhibits greatly improved survival rate without single death over 20 days in comparison with control groups.

Figure S30. Survival rate of the mice for each treatment group (n = 5).

Figure S33. Metabolism of ZNNPs and ZNNPs@FA in different organs. *Ex vivo* real-time NIR-I ($E_x/E_m = 640/720$ nm) and NIR-II ($E_x/E_m = 808/1070$ nm) fluorescence imaging of the mice with i.v. injection of (a) ZNNPs (10 mg/kg) and (b) ZNNPs@FA (10 mg/kg). (c-f) The quantification of above fluorescence images in a and b.

Figure S34. Quantitative analysis of both probes in various organs over time after i.v. injection of

ZNNPs (10 mg/kg) (a) and ZNNPs@FA (10 mg/kg) (b). Error bar indicates SD (n = 2). Every 100 μ L of organ homogenate was pretreated by adding 1 μ L of 50 mM NaHS solution and incubating at 37°C for 2 h.

3. As the author mentioned, the ZNNPs can be used for brain tissue imaging. How did they broken the blood-brain barrier? The authors should give detailed evidence and explanation.

Reply: Thanks for your great questions! As we all known, one of the huge challenges faced for the diagnosis and treatment of brain disease is how the drugs can go through the BBB and accumulate at the focal region effectively. Based on the results in Figure 4i and 4j, we could clearly find that our probe ZNNPs could also light-up the injured region of mouse brain due to the high upregulation of the H₂S level. To further confirm this conclusion, we further repeated these experiments. Both healthy and brain-injured mice were i.v. injected with ZNNPs (10 mg/kg) through tail vein. After 4 h post-injection, the mouse brain was imaged by both fluorescence and PA imaging systems at several time intervals. As shown in Figure S21a, the PA₆₈₀ signals enhanced gradually with the increasing time, while only low PA₉₀₀ signals were recorded over time. In huge contrast, for the healthy mice almost no PA signals at either 680 nm or 900 nm were detected all the time. Moreover, we also conducted *ex vivo* fluorescence imaging of the mouse brain resected from the mice. The results in Figure S21b and S21c shows that the injured tissues in the resected brain can be obviously identified by naked eyes and fluorescence both at 720 nm and 1070 nm. However, no fluorescence was detected from the control mice at all, indicating very little probe has entered the mouse brain. Together, these results strongly demonstrate that probe ZNNPs can go through the BBB and entered the injured brain for visualization of the damaged tissues. In light of these solid evidences, we firmly believe that the probes can solely enter the injured mouse brain but difficult for healthy mice. We speculated that the reason for our nanoprobe easily entering the injured mouse brain is because the BBB of brain-injured mice was disrupted, but the healthy mice retain intact and compact BBB, which has been well demonstrated in numerous previously reported papers. To further verify our hypothesis, the brain tissues both from brain-injured mice and healthy mice were resected and sliced for HE and Nissl staining. Figure S21d and S21e shows that much more nerve cells presented in the injured brain tissues as compared to the normal mice. Besides, obvious breakage and loosen could be clearly observed for brain-injured mice from the HE staining results. Collectively, these results strongly support that our probe is more preferable for detection and tracing of brain damage in living mice.

Figure S21. Comparison of the blood-brain barrier (BBB) permeability between intracerebral hemorrhage (ICH) mice and normal mice (Control). (a) PA images of ICH mice and normal mice with i.v. injection of ZNNPs (10 mg/kg), respectively. (b) and (c) Ex vivo NIR-I (Ex=640 nm, Em=720 nm) and NIR-II (Ex=808 nm, Em=1070 nm) fluorescence imaging of the mouse brains. (d) H&E staining images of ICH mice and normal mice. (e) Nissl staining images of ICH mouse brain, showing injured region by a lack of staining.

The related literature is listed below.

1. Keep, RF, et al. Blood-brain barrier function in intracerebral hemorrhage. In: Cerebral Hemorrhage (eds Zhou L-F, et al.). Acta Neurochir. 105,73-7 (2008). DOI: 10.1007/978-3-211-09469-3_15.
2. Li, Z, et al. Brain transforms natural killer cells that exacerbate brain edema after intracerebral hemorrhage. J Exp Med. 217, (2020). DOI: 10.1084/jem.20200213.
3. Sun, Q, et al. Neurovascular Units and Neural-Glia Networks in Intracerebral Hemorrhage: from Mechanisms to Translation. Transl Stroke Res. 12, 447-460 (2021).
4. Wang, T, et al. Poloxamer-188 can attenuate blood-brain barrier damage to exert neuroprotective effect in mice intracerebral hemorrhage model. J Mol Neurosci. 55, 240-250 (2015).
5. Wu, X, et al. NDP-MSH binding melanocortin-1 receptor ameliorates neuroinflammation and BBB disruption through CREB/Nr4a1/NF-kappaB pathway after intracerebral hemorrhage in mice. J Neuroinflammation. 16, 192 (2019).

4. The NIR-II cellular imaging also should be performed for H₂S response.

Reply: Thanks for your great comments! To evaluate the capability of probe ZNNPs for H₂S detection in living cell, we also carried out the NIR-II imaging (Ex=808 nm, Em=1070 nm) for HCT116 cells after being incubated with ZNNPs for 4 h. As shown in Fig. S16, obvious NIR-II fluorescence was detected in HCT116 cells as control group. However, enhanced NIR-II

fluorescence signals were determined if the cells were pre-incubated with ZnCl₂ (40 μg/mL, 10 min), indicative of lower concentration of H₂S than the control cells. Moreover, if the cells were pre-incubated with extraneous NaHS (100 μM, 1h) or L-Cys (24 μg/mL, 1h) followed by ZNNPs treatment, the fluorescence signals were dramatically suppressed in comparison with the control cells, which is highly consistent with the observed F₁₀₇₀ decreasing for the assays containing probes and NaHS solution.

Figure S16. NIR-II imaging of HCT116 cells treated with ZNNPs (20 μg/mL). (a) NIR-II confocal fluorescence images and (b) quantitative fluorescence intensity of HCT116 cells that were treated with PBS buffer, ZnCl₂ (40 μg/mL, 10 min), NaHS (100 μM, 1 h), and L-Cys (24 μg/mL, 1 h). Values represent mean ± SD (n = 4, *P < 0.05, **P < 0.01, ***P < 0.001).

5. The accumulation of nanoparticles in tumors was limited, which will affect the therapeutic effect.

Reply: I appreciate the reviewer's question. I fully agree with the reviewer. It is well known that all of nanoparticles actually accumulated within tumors mainly based on the EPR effect, and a large number of studies have already demonstrated that less than 10% of total nanoparticles are ultimately enriched in the tumors, which is indeed not sufficient for achieving efficient treatment of tumors. In fact, this problem is one of the huge challenges faced in nanomedicine currently. Thus, researchers are trying to improve the accumulation of nanoparticles in tumors by modifying the surface of nanoparticles with tumor-targeted groups. In our work, the tumor-targeted group---folic acid was decorated onto the surface of ZNNPs to improve the accumulation of nanoprobe in colorectal tumors. Nevertheless, it still far away from the achievement of highly effective tumor treatment. Therefore, our research is mainly for proof of the concept. Hopefully our findings will provide new insight for efficient treatment of colorectal tumors in clinics.

6. What is the detection limitation of the explored nanoprobe for H₂S?

Reply: Thanks for your great comments! To determine the detection limit of ZNNPs for H₂S, 14 μg/mL of ZNNPs was incubated with various concentrations of NaHS ranging from 0 to 500 μM for 10 min followed by detection of the fluorescence (FL₇₂₀) and ratiometric photoacoustic (PAS₆₈₀/PAS₉₀₀) signal. Figure S11c and S11d clearly shows that both FL₇₂₀ and PAS₆₈₀/PAS₉₀₀ were enhanced linearly with the increase of NaHS concentrations. According to the formula LOD = 3σ/s (where LOD was the limit of detection, σ was the standard deviation of blank probe sample measurement, and s was the slope of the calibration pot, values are mean ± SD (n = 3)), the lowest detection limits of H₂S in fluorescence and ratiometric PAS₆₈₀/PAS₉₀₀ were eventually calculated

to be $\sim 4.7 \mu\text{M}$ and $\sim 0.68 \mu\text{M}$, respectively, which again demonstrates that the sensitivity of ratiometric PA is higher than NIR fluorescence imaging in H_2S detection.

7. Authors also missed the QY of nanoprobes.

Reply: Thanks for the reviewer's valuable suggestion! We further measured the fluorescence quantum yields of nanoprobe ZNNPs in PBS buffer (pH 7.4) with and without treatment of NaHS (500 μM). NIR II dye IR-26 ($\Phi = 0.05\%$) and Cy5.5 ($\Phi = 23\%$) was chosen as the references. As shown in Figure S11a, after calculation, the fluorescence quantum yields (QY) of ZNNPs with and without NaHS are 17.2% and 1.1%, respectively, at the excitation wavelength of 660 nm. Besides, the QY of ZNNPs with and without NaHS are 0.006% and 0.1%, respectively, at the excitation wavelength of 808 nm.

	QY(Φ) Ex=660nm	QY(Φ) Ex=808nm
ZNNPs	1.1%	0.1%
ZNNPs+NaHS (500 μM)	17.2%	0.006%

Figure S11a. Fluorescence quantum yields (QY) measurement of ZNNPs in PBS buffer (pH 7.4) with and without incubation of NaHS (500 μM). All the measurements were carried out at room temperature.

8. In Figure S20, there are obvious interference signal from NIR-II imaging and NIR imaging. It seems the specific diagnosis of HCT116 tumor is very poor. And the authors should give the detailed explanation on the interference signal in the activated nanoprobe.

Reply: Thanks for the reviewer's question! I agree with you. It is true that there are obvious non-specific fluorescence signals in both NIR and NIR-II imaging of tumor-bearing mice. We speculate that there are two major reasons: first, as mentioned above, nanoparticles mainly accumulated within tumors based on the well-known EPR effect, only very little amount (less than 10%) of nanoparticles was ultimately enriched within the tumor, a lot of nanoparticles were simultaneously uptake in liver and other organs, which inevitably produces non-specific fluorescence signals. Second, many studies have demonstrated that high concentration of endogenous H_2S ($\sim 20 \mu\text{M}$) presents in mouse liver and colorectal tumor (0.3 to 3.4 mM). By contrast, the concentration of H_2S in tumors is much higher than the one in the liver with up to 10-fold. Besides, our purpose in current work is not only intending to develop a promising probe for diagnosis of colorectal tumor, but also realize noninvasive and quantitative visualization of endogenous H_2S in vivo by taking advantage of the ratiometric PA imaging feature of our probes. Hence, these interference signals in other organs will have little effect on the quantitative visualization of H_2S in tumors.

Reviewer #3 (Remarks to the Author):

Comments: In the article "Quantitatively Visualizing and Depleting Endogenous H₂S Combined with Activated Photodynamic Effect for Synergistical Enhanced Colorectal Cancer Therapy", the authors report a smart, H₂S-responsive and depleting nanoplatfrom (ZNNPs) to quantitatively and real-time imaging endogenous H₂S for early diagnosis and treatment of H₂S-associated diseases (acute hepatotoxicity, cerebral hemorrhage, and colorectal tumor). This ZNNPs nanoplatfrom displayed unexpected NIR conversion (F₁₀₇₀→F₇₂₀) and ratiometric photoacoustic (PA₆₈₀/PA₉₀₀) signal responsiveness towards H₂S and simultaneously depleted the mitochondrial H₂S level in cancer cells leading to remarkable reduction of glycolysis and severe mitochondrial damage, together with the activated photodynamic effect. Sufficient experiments and data were supplied to make this work nice. The nanoplatfrom is new and the application is important. Thus, I recommend its publication after a minor revision.

Response: Thanks for your affirmation and encouragement to us.

1) Reports with similar proposals and structures utilizing different chemistry and imaging mode have been reported in recent years (DOI: 10.1021/jacs.9b09181; 10.1021/jacs.5b04097 etc.). What is the big improvement of this work utilizing H₂S-responsive ZNNPs nanoplatfrom in comparison with those reports? Deeper discussion should be provided to make the manuscript persuasive.

Response: Thanks for the reviewer's valuable comments! The design of our project in this work is actually inspired by these reported literatures. In terms of chemical design, our work is not very novel. We utilized the nucleophilic substitution reaction between 4-nitrobenzoic ester and H₂S to produce the ratiometric FL and PA signals for H₂S detection, which is quite similar to previously reported work (e.g. DOI: 10.1021/jacs.9b09181). By contrast, the maximal FL/PA signals in reported literature are below 800 nm whose limited tissue penetration seriously hampered their wide applications in vivo. However, differing from all of the previous studies, the novelty and innovation in our work include below two aspects: first, the key novelty of our study is that we not only developed a new H₂S-responsive NIR/ratiometric photoacoustic imaging probe for visualization of endogenous H₂S in vivo, more importantly, taking advantage of the ratiometric PA feature, we for the first time established a quantitative equation between ratiometric PA signal and the level of endogenous H₂S. In light of this relationship, we successfully realized the noninvasive and quantitative visualization of endogenous H₂S in mouse liver, brain as well as colorectal tumor. This approach may offer a powerful tool for studying the vital impacts of H₂S in related diseases; Second, we for the first time found that the depletion of the endogenous H₂S in mitochondria with our probe could cause significant ATP reduction and severe mitochondrial damage, combined with the H₂S-activated photodynamic effect, eventually leading to efficient suppression of colorectal tumor in living mice, which would provide a new insight for achieving precise diagnosis and treatment of colorectal cancer.

2) Why author use two different cells (human embryonic kidney 293 (HEK293) cells and HCT116 cells) in this work?

Response: Thanks for the reviewer's question! A large numbers of studies have demonstrated that colorectal cancer cells present high level of H₂S in comparison with normal cells. In our work, there are two reasons we used two different cell lines, namely human embryonic kidney 293

(HEK293) cells and colorectal cancer HCT116 cells. First, to demonstrate the specificity of our probes toward H₂S, we chose HEK293 cells and HCT116 cells as negative control and positive cells for better comparison. Second, to validate whether the depletion of endogenous H₂S can cause the mitochondrion damage leading to effective cancer cell apoptosis, it is better to use one normal cell line and one cancer cell line for comparative study.

3) In Fig. 3 and Fig. 7, please mark the scale bar in fluorescence and PA images.

Response: Thank you for pointing out our mistake! We have added the scale bar in both Figures properly.

4) The singlet oxygen (¹O₂) quantum yield of ZNNPs@FA should be provided.

Response: Thanks for the reviewer's valuable suggestion! We further measured the singlet oxygen (¹O₂) quantum yield of ZNNPs@FA. Methylene blue dissolved in acetonitrile ($\Phi_{\Delta} = 0.52$) was used as the reference. As shown in Figure S27, the ¹O₂ quantum yield of ZNNPs@FA gradually enhanced with the increasing concentrations of NaHS ranging from 0 to 300 μM , implying that the ¹O₂ quantum yield of ZNNPs@FA is positively associated with the concentration of H₂S in a dose-dependent manner.

Figure S27. Measurement of singlet oxygen (¹O₂) quantum yield for probe ZNNPs@FA. Methylene blue in acetonitrile ($\Phi_{\Delta} = 0.52$) was used as the reference. (a) Curve fitting of DPBF consumption rate in methylene blue (30 $\mu\text{g/mL}$) solution. (b-f) Curve fitting of DPBF consumption rate in ZNNPs@FA (30 $\mu\text{g/mL}$) solution with different concentrations of NaHS (0, 37.5, 75, 150, and 300 μM). (g) ¹O₂ quantum yields of ZNNPs@FA in the presence of different concentrations of NaHS.

Reviewers' Comments:

Reviewer #1:

Remarks to the Author:

The authors have collected solid data to well address the comments raised by the reviewers. These new data together with the revised manuscript have improved the quality of the manuscript. As the authors described in the rebuttal letter, they have present the novelty in two aspects, including 1) quantitative measurement of H₂S in living systems through ratiometric imaging and 2) first demonstration of ability to delete mitochondrial H₂S that improving photodynamic therapy of colorectal tumors. In addition, the authors also demonstrated the potential for the imaging of H₂S in injury brain, which has not be reported before. In consideration of the results demonstrated, the paper is suggested to be accepted for the publication in Nature Communications.

Reviewer #2:

Remarks to the Author:

The authors have addressed the most comments. There is still a problem needed to be addressed. The size distribution (Figure S5) revealed the size of ZNNPs and ZNNPs@FA was estimated to 30-100 nm. While, in the in vivo distribution results (Figure S34), both of the ZNNPs and ZNNPs@FA presented high accumulation in the Kidney. As it is well known that only sub 6 nm nanoparticles can be efficiently cleared from kidney. The authors should give detailed explanation for this problem.

Reviewer #3:

Remarks to the Author:

All my previous concerns have been addressed. I suggest that the manuscript should be accepted without further revision.

Reviewer #1 (Remarks to the Author):

Comments: The authors have collected solid data to well address the comments raised by the reviewers. These new data together with the revised manuscript have improved the quality of the manuscript. As the authors described in the rebuttal letter, they have present the novelty in two aspects, including 1) quantitative measurement of H₂S in living systems through ratiometric imaging and 2) first demonstration of ability to delete mitochondrial H₂S that improving photodynamic therapy of colorectal tumors. In addition, the authors also demonstrated the potential for the imaging of H₂S in injury brain, which has not be reported before. In consideration of the results demonstrated, the paper is suggested to be accepted for the publication in Nature Communications.

Response: Thanks for your affirmation and encouragement to us.

Reviewer #2 (Remarks to the Author):

Comments: The authors have addressed the most comments. There is still a problem needed to be addressed. The size distribution (Figure S5) revealed the size of ZNNPs and ZNNPs@FA was estimated to 30-100 nm. While, in the in vivo distribution results (Figure S34), both of the ZNNPs and ZNNPs@FA presented high accumulation in the Kidney. As it is well known that only sub 6 nm nanoparticles can be efficiently cleared from kidney. The authors should give detailed explanation for this problem.

Response: We appreciate the reviewer's comments! It is true that both ZNNPs and ZNNPs@FA presented in buffer as nanoparticles with the size of around 100 nm. In principle, these nanoparticles are difficult to be excreted from the body through the kidney if they are stable enough even in complicated living system. In fact, our nanoprobe were fabricated by encapsulating NIR-II fluorophore ZM1068-NB with PEG polymers, which is not so stable especially in liver. So I speculate that the nanoprobe can be gradually dissociated leading to the accumulation of NIR-II fluorophore in the kidney. This should be the reason we measured the fluorescence signals in the kidney of mice.

Reviewer #3 (Remarks to the Author):

Comments: All my previous concerns have been addressed. I suggest that the manuscript should be accepted without further revision.

Response: Thanks for your affirmation and support to us.